# Visual cue-related activity of cells in the medial entorhinal cortex during navigation in virtual reality

**Amina A Kinkhabwala[1,2,3†‡]\*, Yi Gu[1,2,3†§]\*, Dmitriy Aronov[1,2,3#], David W Tank[1,2,3]\***

[1]Princeton Neuroscience Institute, Princeton University, Princeton, United States; [2]Bezos Center for Neural Circuit Dynamics, Princeton University, Princeton, United States; [3]Department of Molecular Biology, Princeton University, Princeton, United States

**\*For correspondence:**
amina.kinkhabwala@gmail.com (AAK);
guyi.thu@gmail.com (YG);
dwtank@princeton.edu (DWT)

[†]These authors contributed equally to this work

**Present address:** [‡]Department of Biology and Biological Engineering, California Institute of Technology, Pasadena, United States; [§]Spatial Navigation and Memory Unit, National Institute of Neurological Disorders and Stroke, National Institutes of Health, Bethesda, United States; [#]Department of Neuroscience, Zuckerman Mind Brain Behavior Institute, Columbia University, New York, United States

**Competing interests:** The authors declare that no competing interests exist.

**Abstract** During spatial navigation, animals use self-motion to estimate positions through path integration. However, estimation errors accumulate over time and it is unclear how they are corrected. Here we report a new cell class ('cue cell') encoding visual cues that could be used to correct errors in path integration in mouse medial entorhinal cortex (MEC). During virtual navigation, individual cue cells exhibited firing fields only near visual cues and their population response formed sequences repeated at each cue. These cells consistently responded to cues across multiple environments. On a track with cues on left and right sides, most cue cells only responded to cues on one side. During navigation in a real arena, they showed spatially stable activity and accounted for 32% of unidentified, spatially stable MEC cells. These cue cell properties demonstrate that the MEC contains a code representing spatial landmarks, which could be important for error correction during path integration.

## Introduction

Animals navigate using landmarks, objects or features that provide sensory cues, to estimate spatial location. When sensory cues defining position are either absent or unreliable during navigation, many animals can use self-motion to update internal representations of location through path integration (*Mittelstaedt, 1982*; *Tsoar et al., 2011*). A set of interacting brain regions, including the entorhinal cortex, parietal cortex, and the hippocampus (*Brun et al., 2008*; *Bush et al., 2015*; *Calton et al., 2003*; *Calton et al., 2008*; *Clark et al., 2010*; *Clark et al., 2013*; *Clark et al., 2009*; *Clark and Taube, 2009*; *Frohardt et al., 2006*; *Geva-Sagiv et al., 2015*; *Golob and Taube, 1999*; *Golob et al., 1998*; *Hollup et al., 2001*; *Moser et al., 1993*; *Parron et al., 2004*; *Parron and Save, 2004*; *Taube et al., 1992*; *Whitlock et al., 2008*) participate in this process.

The MEC is of particular interest in path integration. Grid cells in the MEC have multiple firing fields arrayed in a triangular lattice that tile an environment (*Hafting et al., 2005*). This firing pattern is observed across different environments with the grid cell population activity coherently shifting during locomotion (*Fyhn et al., 2007*). These observations have led to the hypothesis that grid cells form a spatial metric used by a path integrator. Given this, theoretical studies have demonstrated how velocity-encoding inputs to grid cell circuits could shift grid cell firing patterns, as expected of a path integrator (*Barry and Burgess, 2014*; *Burak and Fiete, 2009*; *Fuhs and Touretzky, 2006*; *McNaughton et al., 2006*). Cells encoding the speed of locomotion have been identified in this region (*Kropff et al., 2015*), providing evidence of velocity-encoding inputs and further support for the role of MEC in path integration.

A general problem with path integration is the accumulation of errors over time. A solution to this problem is to use reliable spatial cues to correct estimates of position (*Evans et al., 2016*;

*Hardcastle et al., 2015*; *Pollock et al., 2018*). Many recent experimental studies showed profound impairment of grid cell activity by altering spatial cues, including landmarks and environmental boundaries. For example, the absence of visual landmarks significantly disrupted grid cell firing patterns (*Chen et al., 2016*; *Pérez-Escobar et al., 2016*). Also, experiments that maintained the boundaries of a one-dimensional environment but manipulated nonmetric visual cues caused rate changes in grid cells (*Pérez-Escobar et al., 2016*). The decoupling of an animal's self-motion and visual scene altered grid cell firing patterns (*Campbell et al., 2018*). Finally, many studies have shown that grid cell firing patterns were influenced by nearby boundaries (*Carpenter et al., 2015*; *Derdikman et al., 2009*; *Giocomo, 2016*; *Hardcastle et al., 2015*; *Krupic et al., 2015*; *Krupic et al., 2018*; *Stensola et al., 2015*; *Yamahachi et al., 2013*).

Border cells in the MEC, with firing fields extending across environmental boundaries (*Solstad et al., 2008*), are good candidates for supplying information for error correction near the perimeter of simple arenas (*Pollock et al., 2018*). This role of border cells is supported by the fact that an animal's interactions with boundaries yielded direction-dependent error correction (*Hardcastle et al., 2015*). However, grid cell firing fields are maintained throughout open arenas in locations where border cells are not active and thus cannot participate in error correction. It is possible that cue cells, like border cells, could provide a mechanism for error correction.

Also, natural navigation involves moving through landmark-rich environments with higher complexity than arenas with simple boundaries. How information from a landmark-rich environment is represented within the MEC is unknown. If there were cells in the MEC that encoded sensory information of landmarks, then more robust path integration and error correction of grid cells would be possible using circuitry self-contained within this brain area. In the MEC, while border cells have been shown to respond to landmarks in virtual reality (*Campbell et al., 2018*), increasing evidence suggests that unclassified cells also contain information about spatial environments (*Diehl et al., 2017*; *Hardcastle et al., 2017*; *Høydal et al., 2019*). It would be useful to further determine whether these unclassified cells represent spatial cues (other than borders) that could be used in error correction.

Here we address this question by recording from populations of cells in the MEC during virtual navigation along landmark-rich linear tracks using electrophysiological and two-photon imaging approaches. Virtual reality (VR) allows for complete control over the spatial information of the environment, including the presence of spatial cues along the track. The animal's orientation within the environment was also controlled, simplifying analysis. We report that a significant fraction of previously unclassified cells in MEC responded reliably to prominent spatial cues. As a population, the cells fired in a sequence as a spatial cue is passed. When some cues were removed, cue cell firing fields near those cues were no longer present. When cues were presented on either the left or right side of the track, these cells subdivided into left and right-side preferring cue cells. During navigation along different virtual tracks, cue cells largely maintained the same cue-aligned firing patterns. These cells could provide position information necessary in local MEC circuits for error correction during path integration in sensory rich environments, which are regularly found in nature.

## Results

### Cue-responsive cells in virtual reality

Mice were trained to unidirectionally navigate along linear tracks in virtual reality to receive water rewards. Virtual tracks were 8 meters long and had a similar general organization and appearance: tracks began with a set of black walls, followed by a short segment with patterned walls, and then finally the majority of the track was comprised of a long corridor with a simple wall pattern (*Figure 1A*). Different environments were defined by distinct pairs of identical visual cues (tower-like structures) symmetrically present along both sides of the corridor. These cues were non-uniformly spaced along the track and the last cue was always associated with a water reward.

We used tetrodes to record 789 units in the MEC of three mice (Materials and methods and *Figure 1—figure supplement 1*). Activity of a subpopulation of these units exhibited a striking pattern, with spiking occurring only near cue locations along the virtual linear tracks (*Figure 1A*). On each run along the track, clusters of spikes were present at cue locations, forming a vertical band of spikes at each cue in the run-by-run raster plot. Spatial firing rates were calculated by averaging this spiking

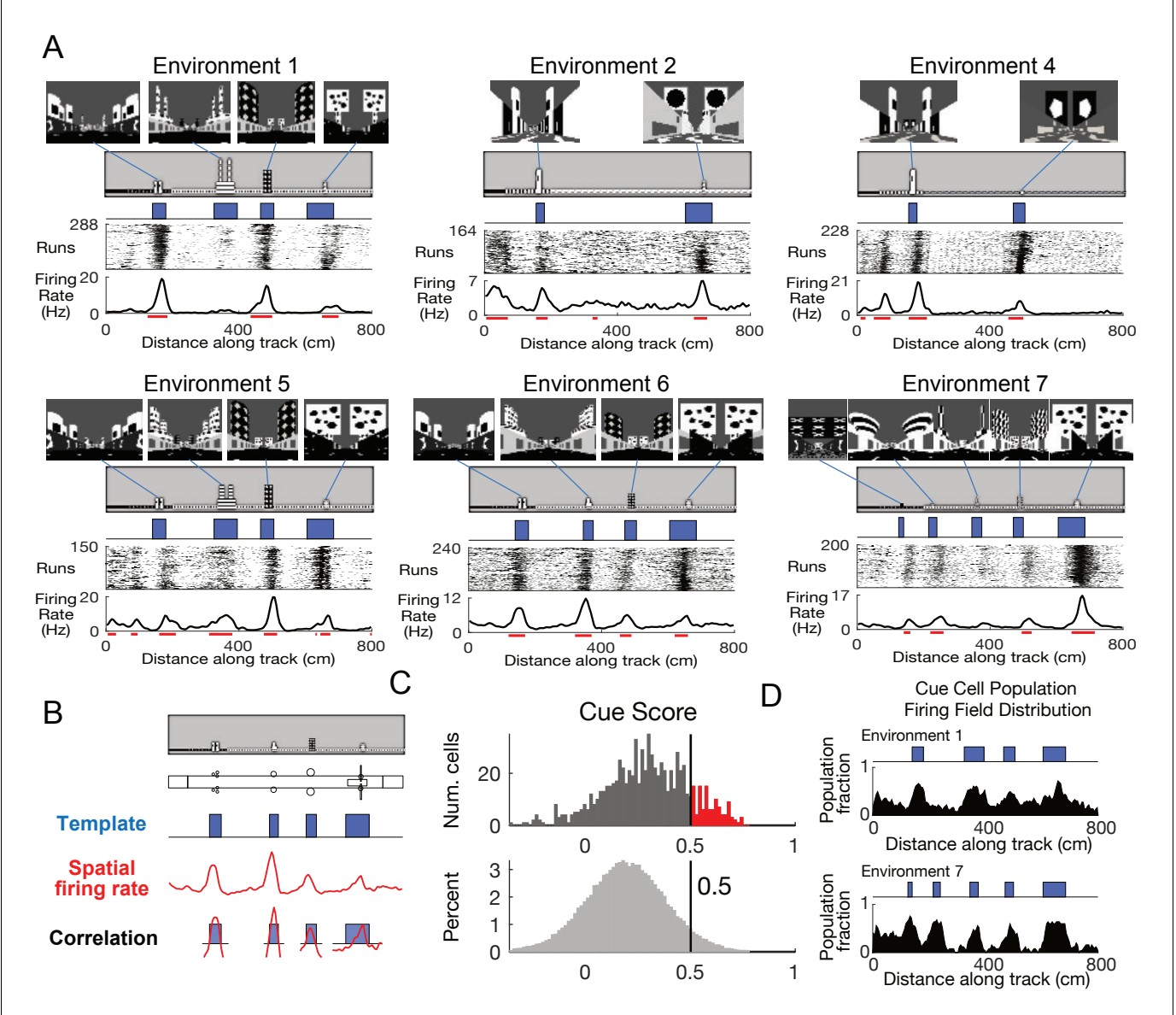

**Figure 1.** Cells respond to cues in a virtual linear environment. (**A**) Examples of cells with cue-related activity recorded during navigation along virtual tracks. At the top of each example are views of each cue from the animal's perspective inside the track at that location. Side views of the track are shown below, with the start location to the left. The raster plot for a single cell's spatial activity pattern across multiple traversals of the track is plotted with the average firing rate (Hz) as a function of track position (spatial firing rate) below. Spatial firing fields for the cell are indicated with horizontal red bars. (**B**) Calculation of cue score. The Pearson correlation between the cue template and the cell's spatial firing rate was calculated and the spatial shift was defined as the local maximum closest to zero. The cue template was translated by this shift and the correlations of this shifted cue template and spatial firing rate at each cue were individually calculated. The cue score was defined to be the mean of these correlations (Materials and Methods). (**C**) The distribution of cue scores of recorded cells is shown at the top with the distribution of cue scores calculated on shuffled data shown below. The threshold was chosen as the value that 95% of the shuffled scores did not exceed (vertical black line). Cells exceeding this threshold were termed 'cue cells' and are shown in red in the top plot. (**D**) Distribution of spatial firing fields of all cue cells in two environments.

The online version of this article includes the following source data and figure supplement(s) for figure 1:

**Source data 1.** Cue score and shuffle distributions.
**Source data 2.** Cue cell population field distributions across the virtual tracks.
**Figure supplement 1.** Histology of tetrode tracks and tetrode cell type summary.
**Figure supplement 2.** Velocity profiles of navigation along virtual tracks.
**Figure supplement 3.** Cue cells in tetrode lowering database.

activity across all runs along the track. Prominent peaks in the spatial firing rate were present at cue locations. In order to classify these peaks, we defined spatial firing fields as the locations along the track where the spatial firing rate exceeded 70% of the shuffled data (Materials and Methods) and observed that the spatial firing fields were preferentially located near cue locations (fields indicated by red lines in *Figure 1A*).

To quantify this feature of the spatial firing rate, we developed a 'cue score' that measures the relationship between a cell's spatial firing rate and the visual cues of the environment (*Figure 1B* and Materials and Methods). The cue score was based on the correlation of the cell's spatial tuning with a spatial cue template that had value one at each cue, and zero elsewhere. Cells with cue scores above the threshold (95th percentile of shuffled data, Materials and Methods) represented ~13% of all recorded cells (*Figure 1C*). In the remainder of the paper, we refer to these cells as 'cue cells'.

We next quantified the distribution of spatial firing fields of all cue cells along the track by calculating, for each 5 cm bin, the fraction of cue cells with a spatial firing field in that bin (Materials and methods). We defined the plot of this fraction versus location as the firing field distribution for all cue cells. This distribution had peaks in locations where salient information about the environment was present (*Figure 1D*), and some fields were correlated with the beginning of the track where wall patterns changed. The mean firing field fraction for spatial bins of the firing field distribution in cue regions ($0.4 \pm 0.1$) was higher than that for bins outside of cue regions ($0.2 \pm 0.1$) (paired one-tailed t-test: fraction of cue cell population with fields located in cue region bins > fraction of cell population with fields in bins outside cue regions, N = 100, 3 animals, $p=1.2 \times 10^{-11}$). Thus, the cue score identifies a subpopulation of MEC cells with spatial firing fields correlated with prominent spatial landmarks.

## Cue cell responses to environment perturbations

Is the activity of cue cells truly driven by the visual cues of the environment? To address this question, we designed related pairs of virtual tracks. One track had all cues present (*with-cues* track) and, in the second track, the last three cues were removed (*missing-cues* track). Tetrode recordings were performed as mice ran along both types of track in blocks of trials within the same session. Water rewards were delivered in the same location on each track regardless of the cue location differences.

At the beginning of both *with-cues* and *missing-cues* tracks, where the tracks were identical, the spatial firing rates of cue cells were similar across tracks. Vertical bands of spikes were present in the run-by-run raster plots of both tracks and formed peaks in the spatial firing rate. The bands were also identified as spatial firing fields, which generally aligned to features of the environment (spatial cues/changes in wall patterns) present on both tracks (*Figure 2A*, red lines indicate field locations). However, the firing patterns changed dramatically from the point along the track where the environments began to differ. Spatial firing fields were prominent at cue locations along the entire remaining part of the *with-cues* track (*Figure 2A*, top) but were not present on the same part of the *missing-cues* track (*Figure 2A*, bottom). To quantify this difference, we performed two calculations using an equal number of runs for both the *with-cues* and *missing-cues* tracks (*Figure 2B and D*). In both cases, the data were split into two regions along the track (top of *Figure 2B and D*): the start region where cues were present for both tracks (black bar marks this region, Region A - same) and the rest of the track where cues were either present or absent (green bar, Region B - different). We first calculated the Pearson correlation with lags varying from −300 to 300 cm in 5 cm steps. The correlation between the firing rate and template was defined to be the value of the correlation at the peak in the Pearson correlation located closest to zero shift. This correlation was lower for the firing rates on the *missing cues* track in Region B but not for region A (*Figure 2B*, paired one-tailed t-test: correlation in Region B on *with-cues* track > correlation in Region B on *missing-cues* track, N = 65, 3 animals, for region A: p=0.56; for region B: $p=1 \times 10^{-14}$). We also compared changes of the spatial firing field distribution of the cue cell population (*Figure 2C*) across the two tracks and, for each cue cell, the fraction of the track region containing spatial firing fields (*Figure 2D*). In *Figure 2C*, for each 5 cm bin along the track, the fraction of the cue cell population with a field in that bin are plotted for the *with-cues* and *missing-cues* tracks for two environments. In Region A for both tracks, cue cells showed a similar fraction of the region with fields. Cue cells had spatial firing fields clustered in each region where a cue was located on both tracks, but these fields were not present when cues were removed in Region B of the *missing-cues* track (*Figure 2C*). In Region B, the

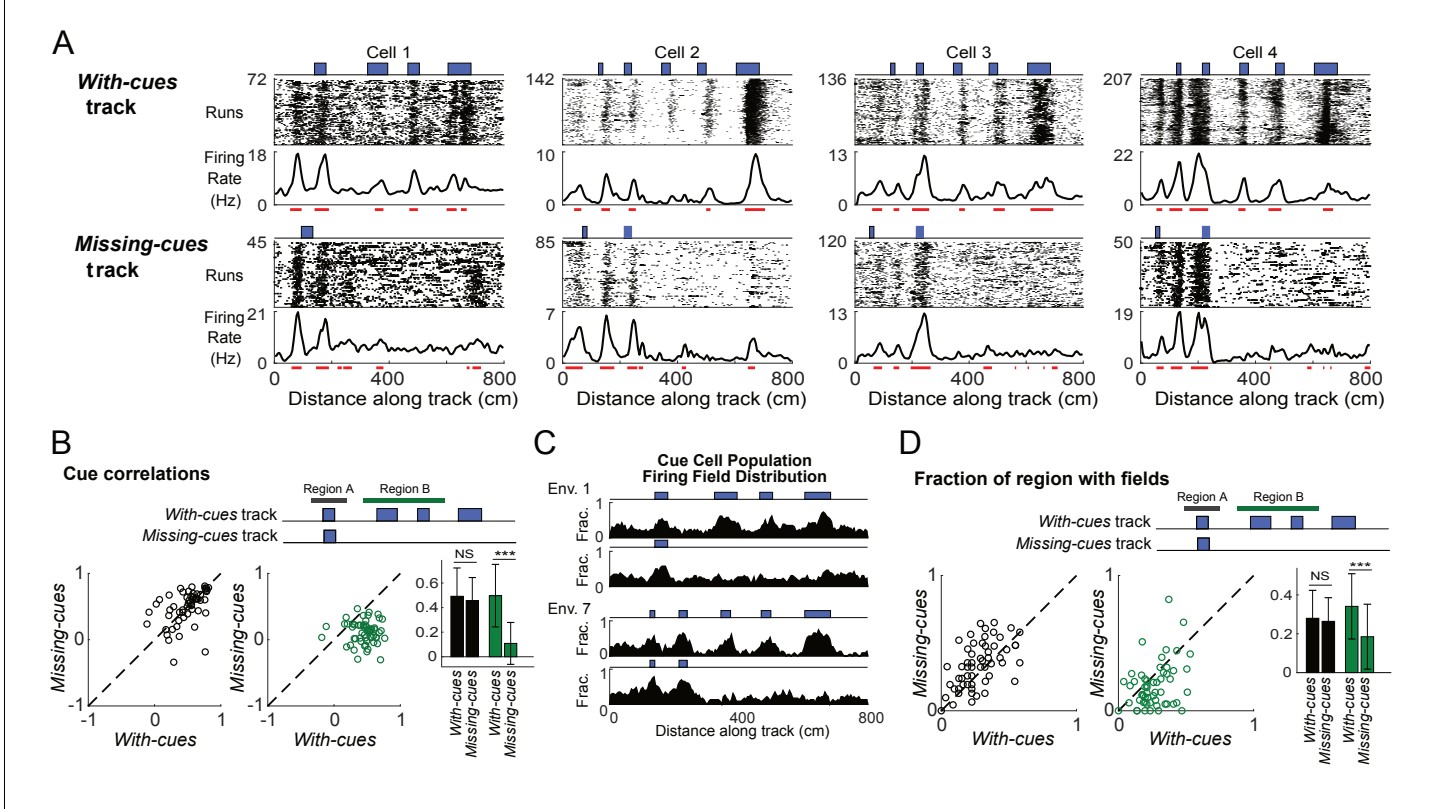

**Figure 2.** Cells respond to cue changes in an environment. (**A**) Examples of the spatial firing rates of cells during cue perturbation experiments. For each example, the top and bottom panels are from the same cell in blocks of trials in which the animal either ran down a virtual track with all cues present (*with-cues* track, top) or a track where some cues were missing in the later part of the track (*missing-cues* track, bottom). The environment and cue template for both environments are shown with the corresponding raster plots and spatial firing rates below. (**B**) Cue correlations of the firing rates along the *with-cues* and *missing cues* tracks within Regions A and B were calculated. The correlation of firing rate to cue was lower for the firing rates on the *missing cues* track in Region B (paired one-tailed t-test: cue correlation on *with-cues* track >cue correlation on *missing-cues* track, N = 65, 3 animals, for Region A, p=0.56, for Region B, p=1×10⁻¹⁴). On the right, the means with standard deviations are shown for regions A and B on each track. (**C**) Population field distribution for the entire population of cue cells along *with-cues* and *missing-cues* tracks for two environments. (**D**) Comparison of firing fields of all cue cells between runs in the initial region that is the same for both tracks (Region A) and the later region (Region B) on the *with-cues* and *missing-cues* tracks. The fraction of each region that had spatial firing fields (number of field bins/total bins in that region) is plotted for each cue cell. The field fraction was larger in region B on the *with-cues* track in comparison to region B on the *missing-cues* track (paired one-tailed t-test: field fraction on *with-cues* track >cue field fraction on *missing-cues* track, N = 65, 3 animals, for region A: p=1.00, for region B, p=0.0002). On the right, the means with standard deviations are shown for regions A and B on each track. ***p≤0.001. Student's t-test.

The online version of this article includes the following source data and figure supplement(s) for figure 2:

**Source data 1.** Cue cell firing field fractions.
**Source data 2.** Correlations of firing rates along tracks to cue template.
**Figure supplement 1.** Cue-removal responses of cue cells with fields near or far from cues.
**Figure supplement 2.** Cue cell responses to cue changes for tetrode lowering database.

fraction of the region with spatial firing fields was lower in the *missing-cues* track compared to the *with-cues* track (*Figure 2D*, paired one-tailed t-test: field fraction on *with-cues* track > field fraction on *missing-cues* track, N = 65, 3 animals, for region A: p=1.00, for region B: p=0.0002). These results were consistent for both cue cells with varying spatial shifts relative to the cues (cue cells separated into two categories: those with fields either near cues or far from cues, *Figure 2—figure supplement 1*) and for a database consisting of dates in which tetrodes were lowered (*Figure 2—figure supplement 2*). These results demonstrate that cue cells are more coherently active in regions of an environment where cues are located.

## Relationship to previously defined cell classes

To relate our population of cells recorded along linear tracks in virtual reality to previously characterized cell types in the MEC, the same cells were also recorded as the animal foraged for chocolate chunks in a real two-dimensional (2D) environment (0.5m × 0.5 m). From the recordings performed in the real arena, we calculated grid, border, and head direction scores for all cells (i.e. both cue and non-cue cells, Materials and Methods). We plotted the values of these spatial scores against the cue scores, which were calculated for the same cell during VR navigation, to determine the relationship

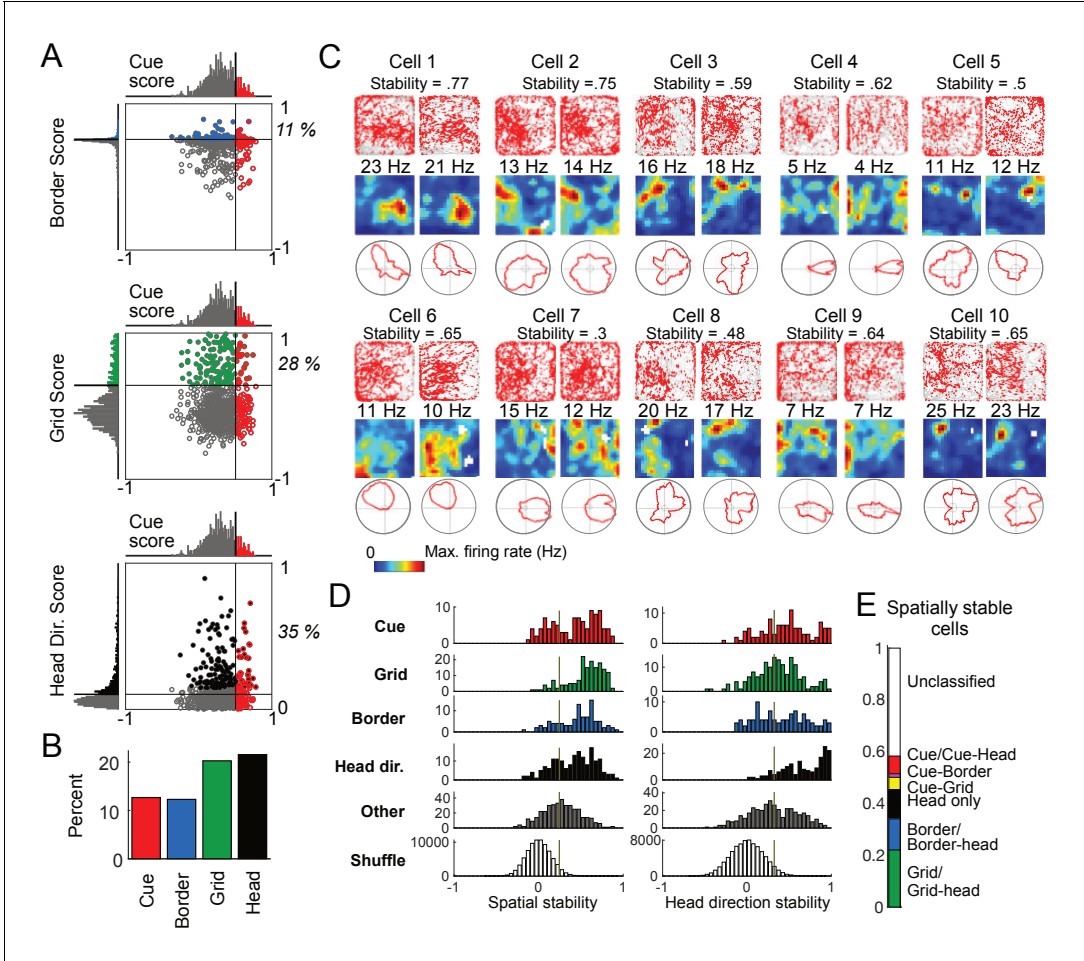

**Figure 3.** Cue cell activity during foraging in a real arena. (A) Relative distributions of cue scores compared to border, grid, and head direction scores. Thresholds were calculated as the value that exceeds 95% of the shuffled scores. The solid line indicates the threshold for each score that was used to determine the corresponding cell type (Materials and Methods). Cells are color-coded for whether they are cue (red), grid (green), border (blue), or head direction cells (black). The percentage of the cue cell population that was conjunctive for border, grid, and head direction is shown in each plot. (B) Percentage of each cell type in the dataset. (C) Examples of the spatial stability of the spatial firing rates of cue cells in a real arena. The recording of each cell was divided in half. The spatial firing rates of the first and second halves are shown for each cell in the left and right columns. Within each column: top: plots of spike locations (red dots) and trajectory (gray lines); middle: the 2D spatial firing rate (represented in a heat map with the maximum firing rate indicated above); bottom: head direction firing rate. The stability was calculated as the correlation of these two firing rates and shown at the top for each cell. (D) Histograms of the spatial and head direction stability of the 2D real environment firing rates by cell type. (E) Percentage of 2D real environment stable cells that are of a certain type. Cell types are color-coded: red = cue cell, green = grid cell, blue = border cell, black = head direction cell.

The online version of this article includes the following source data and figure supplement(s) for figure 3:

**Source data 1.** Cue, border, grid and head direction scores.
**Source data 2.** Spatial and head direction stability values by cell type.
**Figure supplement 1.** Cue cell activity in real arenas.
**Figure supplement 2.** Real arena navigation for tetrode lowering database.

between cue cells and previously defined cell classes (*Figure 3A*). We found that a small percentage of cue cells were conjunctive with border (11%) or grid (28%) cell types, and some cue cells had a significant head direction score (35%) (since the head direction score is based on orientation tuning, we do not consider the head direction cell type to be a spatial cell type). The total percentage of cue cells (13%) in the dataset was comparable to that of grid and border cells (*Figure 3B*).

Since most cue cells (63%) were not conjunctive with a previously known spatial cell type, we next examined their spatial activity patterns during navigation in the real arena. As expected from their scores, most cells had irregular activity patterns in the arena and were not classified as any previously identified spatial cell type (*Figure 3C* and *Figure 3—figure supplement 1*).

One striking feature of the spatial firing patterns of cue cells observed in real environments was the spatial stability of these complex and irregular patterns (Materials and methods). The spatial firing rates in the real arena from the first and the second halves of the recording for 10 cue cells are shown in *Figure 3C*. We calculated the spatial stability as the correlation between these two halves and found that the spatial firing patterns were irregular but surprisingly stable. To further quantify this observation, we calculated the distributions of the stability of both the spatial and head direction firing rates for all cell types (Materials and methods) (*Boccara et al., 2010*). We found that the distributions of stability for unclassified cells (not classified as cue, grid, border, or head direction cells, and labeled as 'Other') were generally shifted towards lower values compared to all the currently classified cells, indicating that a large fraction of the remaining unclassified cells do not stably encode spatial and head direction information in the real arena (*Figure 3D*). In contrast, our newly classified cue cells showed comparable spatial stability as other existing cell types, supporting its ability to encode spatial information during navigation. *Figure 3E* shows the fractions of all cells with significant stability scores for their spatial firing rates in the real arena, classified by cell type. While some cue cells were conjunctive with border or grid cells, a large percentage (63%) of cue cells were previously unclassified as a particular spatial cell type. Cue cells accounted for 7% of the population of spatially stable cells, and for 14% of the previously unclassified spatially stable cells.

## Cue cells form a sequence at cue locations

For many cue cells, we observed that their firing fields had varying spatial shifts relative to the cue locations on virtual tracks. We took all cue cells identified for each virtual track and ordered their spatial firing rates and fields by the values of their spatial shift relative to the cue template, which was the smallest displacement of the cue template to best align with the firing rate (*Figure 4A*; Materials and Methods). We found a striking pattern where cue cells formed a sequence of spatial firing fields that was repeated at each cue. To examine if this pattern was produced by the concentration of neural firing around cues, rather than the alignment and ordering of the data alone, we compared this pattern to that of field-shuffled data, which were created by permuting spatial fields within each spatial firing rate and then ordering these new field-shuffled spatial firing rates in the same manner as for the original spatial firing rates (Materials and Methods). Spatial field shuffled data did not exhibit an obvious sequence (*Figure 4A*, bottom). This difference between the cue cells and shuffled data was further quantified by a ridge-to-background ratio (Materials and Methods), which was computed as the mean firing rate in a band centered on the sequential spatial firing rates of the cue cell population divided by the mean background rate outside of this band. We note that although ordering the spatial firing rates of the cells by their spatial shift was expected to create a ridge of firing rate along the diagonal, the mean ridge/background ratio for cue cells (1.8) was higher than that for spatial field shuffled data ($1.2 \pm 0.35$, $p=9.8 \times 10^{-7}$; N = 100 cue cells, *Figure 4B*). Most cue cells had small spatial shifts (*Figure 4C*). Thus, the sequence represents sequential neural activity preferentially located near cue locations, rather than an artifact of ordering the data.

## Side-preference of cue cells in superficial layers of the MEC

Since cues were bilaterally present along all the tracks studied thus far, it is unclear if cue cells were primarily responding to cues on only one side. To address this, we designed a 10-meter virtual track with asymmetric cues on the left and right sides (*Figure 5A*) and performed cellular-resolution two-photon imaging in order to simultaneously record the responses of a large number of cue cells in the MEC (*Low et al., 2014*). Calcium dynamics of layer 2 cells were specifically measured using the

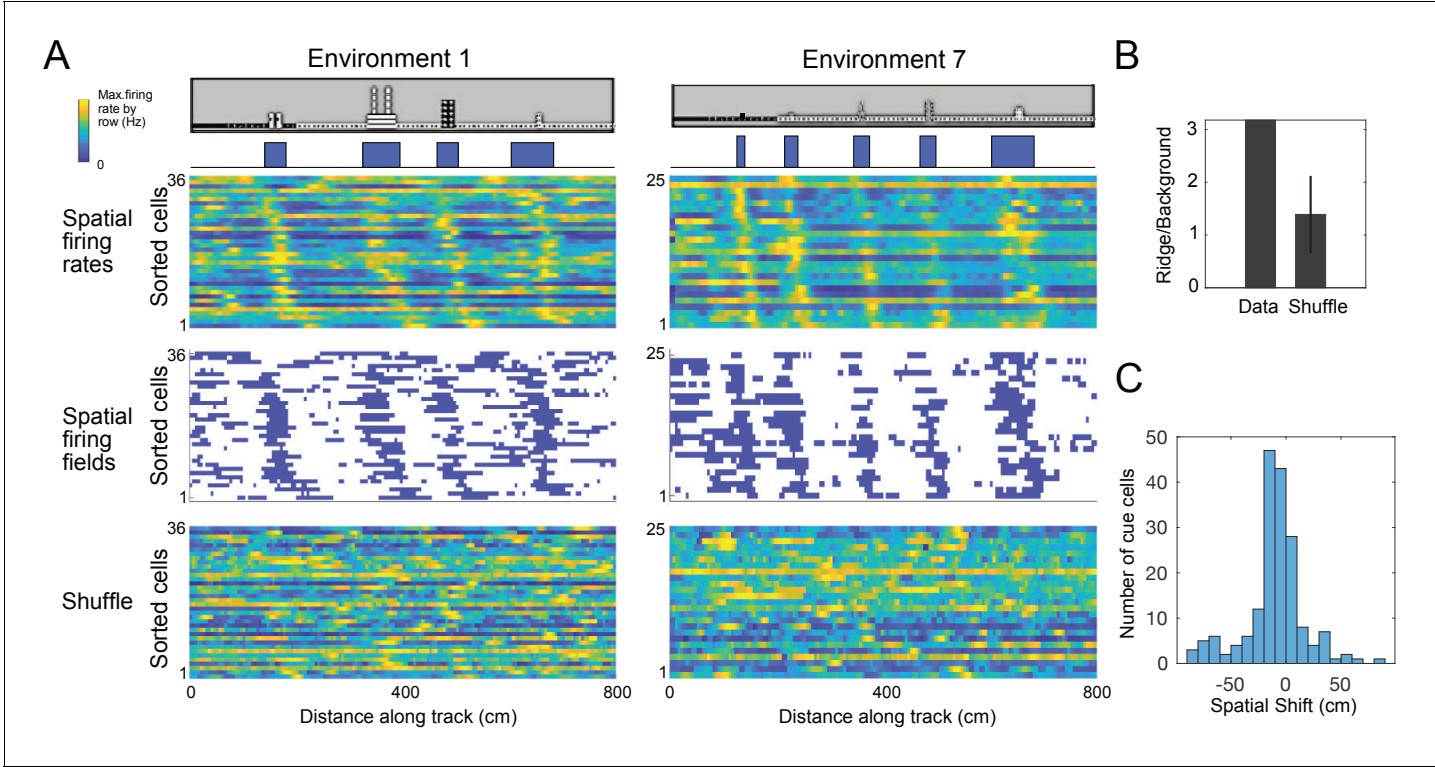

**Figure 4.** Cue cells form a sequence aligned at each cue. (**A**) Cue cell sequences shown for two environments. Top two rows show a side view of the virtual track and the corresponding cue template below. Just below this, the spatial firing rates, where the normalized firing rate of a cue cell is plotted along each row is shown. The cells are sorted based on their spatial shifts calculated for alignment of spatial firing rates to the cue template (Materials and Methods). The corresponding spatial firing fields of the same cells above are shown in the middle panel with the firing fields ordered in the same sequence as the firing rates. At the bottom, an example of the sorted shuffled spatial firing rates which were generated by shuffling the firing fields of each cell and then sorting based on their spatial shifts to the cue templates. (**B**) The ridge to background ratio for the data and shuffles of environment 1. (**C**) Distribution of spatial shifts for all cue cells.

The online version of this article includes the following source data and figure supplement(s) for figure 4:

**Source data 1.** Cue cell sequences.
**Figure supplement 1.** Cue cell sequences in tetrode lowering database.

genetically encoded calcium indicator GCaMP6f, which was stably expressed in layer 2 excitatory neurons of the MEC in GP5.3 transgenic mice (*Figure 5—figure supplement 1A*; *Chen et al., 2013*; *Dana et al., 2014*; *Gu et al., 2018*).

Side-specific cue templates were defined for this environment (*Figure 5B* for left and right cue templates). We classified side-specific cue cells by using the threshold for each template, which was the 95th percentile of shuffled scores obtained using templates with randomly arranged cues (Material and Methods, *Figure 5C and D*). We found that in layer 2 of the left MEC most cue cells only passed the threshold for either left or right template. There were 8.1% and 21.9% of cells uniquely identified as left and right cue cells, respectively. Very few cells passed the thresholds of both templates (14 out of the total population of 281 left and right cells (5%), and 1.6% of the population of all cells, N = 4 animals, *Figure 5—figure supplement 2A*) and their responses correlated to the two single side cue templates under different spatial shifts (*Figure 5—figure supplement 2B*), indicating that they did not simultaneously respond to both left and right cues. Moreover, the cells identified using the template containing both left and right cues had significantly lower cue scores than the left and right cue cells, further suggesting that the cues on both sides were less well correlated than cues on one side (*Figure 5—figure supplement 2C-2G*). Therefore, we only focused on the left and right cue cells in the following analyses.

To directly test whether these left and right cue cells preferentially responded to cues on one side of the track we developed a bilateral score that tested against the null hypothesis that cue cells

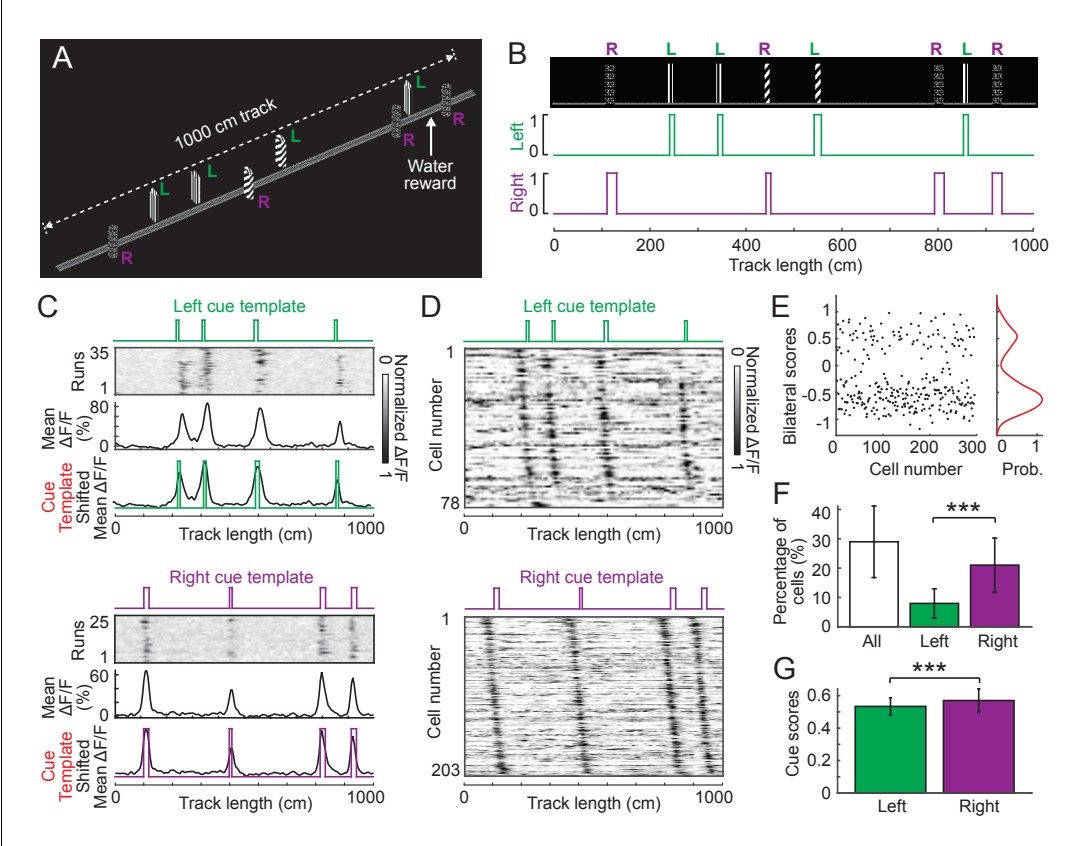

**Figure 5.** Cue cell responses to side-specific cues in layer 2 of the MEC. (**A**) A 1000 cm (10 meter) long virtual linear track for imaging experiments. 'L' and 'R' indicate cues on the left and right sides of the track, respectively. (**B**) Left and right cue templates with cues on the left and right sides of the track. (**C**) Examples of individual cue cells responding to the left- (top) or right-side cues (bottom) in layer 2 of the MEC. For each cell: top: ΔF/F versus linear track position for a set of sequential traversals. Middle: mean ΔF/F versus linear track position. Bottom: overlay of the cue template and aligned mean ΔF/F (black) according to the spatial shift, which gave the highest correlation between them (Materials and Methods). (**D**) Left and right cue cell sequences aligned to left- (top) and right-side cues (bottom), respectively. In each row the mean ΔF/F of a single cell along the track, normalized by its maximum, is plotted. The cells are sorted by the spatial shifts identified from the correlationof their mean ΔF/F to the cue template. (**E**) Bilateral scores of all left and right cells in D. Left: bilateral scores of individual cells (dots). Right: kernel density distribution of bilateral scores. Note that the bilateral scores show a strong bimodal distribution. (**F**) Percentages of cue cells among all cells activeduring virtual navigation (active cells were determined as cells identified using independent component analysis, Materials and Methods) . Left bar: all left and right cue cells. Middle bar: left cue cells. Right bar: right cue cells. Individual data points that were pooled for this are the percentages of cue cells in 12 FOVs in layer 2, p=6.90 × 10$^{-4}$. (**G**) Comparison of cue scores of left and right cue cells in layer 2. Individual data points are cue scores of cells in D,p=1.67 × 10$^{-5}$. All data were generated using layer 2 cue cells in 12 FOVs in four mice. ***p≤0.001. Student's t-test.

The online version of this article includes the following source data and figure supplement(s) for figure 5:

**Source data 1.** Cue scores, bilateral scores and percentages of left and right cue cells.
**Figure supplement 1.** Expression of GCaMP6f in layers 2 and 3 of the mouse MEC.
**Figure supplement 2.** Cue cells preferentially represent cues on a single side, rather than both sides of the track.
**Figure supplement 2—source data 1.** Cue scores, spatial shifts and both-side cue template.
**Figure supplement 3.** Comparison of the percentage of cue cells identified using original and randomized cue templates.
**Figure supplement 4.** Cue cell properties in layers 2 and 3 of the MEC across different environments.
**Figure supplement 4—source data 1.** Cue scores, bilateral scores and percentages of cue cells in layers 2 and 3 on a 18-meter virtual linear track.
**Figure supplement 5.** Spatial shifts of cells with cue-correlated activity patterns.

equally responded to cues on both sides. The bilateral score was defined as the difference between the left and right cue scores of a cue cell when its response was best aligned to its preferred template (*Figure 5—figure supplement 2H*). A large absolute value of the bilateral score (large difference between left and right cue scores) indicated the cell's preferential response to cues on a single side (left or right), whereas a bilateral score around zero (small difference between left and right cue

scores) indicated a comparable response to cues on both sides. We found that the distribution of bilateral scores of left and right cells together was bimodal with two major peaks at large absolute values (*Figure 5E*), suggesting that the left and right cue cells indeed responded to cues on one side of the track.

We next asked whether the locations of left and right cues were preferentially represented in the MEC in comparison to other locations of the track. We compared the percentages of left and right cue cells identified using the current cue templates to those using random templates, which were created by shuffling cue locations along the track. We reasoned that an unbiased representation of all locations along the track should lead to the classification of similar numbers of cue cells regardless of templates. In contrast, we found that the percentages of left and right cue cells corresponding to the current templates were significantly higher than those to random templates (*Figure 5—figure supplement 3*), indicating that the locations of left and right cues were preferentially represented by cue cells. For this reason, cue cells area unique population with spatial fields clustered cue locations, rather than a subpopulation among cells with spatial fields uniformly spanning the environment.

As observed for tetrode-recorded cue cells (*Figure 4A*), the calcium responses of imaged left and right cue cells also had consistent spatial shifts to individual left and right cues, respectively, and the response together formed sequences (*Figure 5D*). In the left MEC, there were more right cue cells than left cue cells (*Figure 5F*) and the cue scores of right cue cells were higher than those of left cue cells (*Figure 5G*).

The above results for cells in layer 2, including the preferred representation of single-side cues, the consistently shifted responses to individual cues, the greater fraction of right cue cells, were also observed for cells in layer 3 of the left MEC when mice navigated on a 18-meter track (*Figure 5—figure supplement 1B* for the specific labeling of layer 3 cells using virus and *Figure 5—figure supplement 4* for features of cue cells). The 18-meter track, which contained a larger number of cues, also allowed us to further validate that responding to individual cues at consistent spatial shifts was a feature of most cells with cue-correlated activity, rather than an artifact of the cue cell selection procedure (see *Figure 5—figure supplement 5* for details, N = 6 animals for layer 2 data, N = 2 animals for layer 3 data).

These data together support the fact that in the superficial layers of the left MEC, cue cells largely responded to cues on one side, while the right cues were preferentially represented over left cues. The responses of both left and right cue cells formed consistent sequences around individual cues on their preferred side.

## Cue cells represent visual cues in different environments

We next asked whether cue cells are a specialized functional cell type that represents cue locations across environments. To investigate this, we measured calcium responses of the same neurons in layer 2 of the left MEC during navigation along two different virtual tracks. We found that many cue cells maintained cue-correlated responses along both tracks (*Figure 6A*, N = 5 animals). In general, the percentages of cue cells and non-cue cells that remained as the same cell types on two different tracks were significantly higher than chance (*Figure 6B*). Finally, cue cells showed highly correlated spatial shifts relative to cue templates across different tracks (*Figure 6C*). These observations suggest that cue cells are a functional cell type representing visual cue information across different environments.

## Discussion

We have described a novel class of cells in MEC—termed cue cells—that were defined by a spatial firing pattern consisting of spatial firing fields located near prominent visual landmarks. When navigating a cue-rich virtual reality linear track, the population of cue cells formed a sequence of neural activity that was repeated at every landmark. When cues were removed, these cells no longer exhibited a sequence of spatial firing fields. This work shows that visual inputs drive cue cells, as additionally indicated by the preferred representation of contralateral cues during navigation along asymmetric tracks. Cue cells maintained cue-related spatial firing patterns across multiple environments, further supporting the hypothesis that cue cells are a distinct functional cell type in MEC.

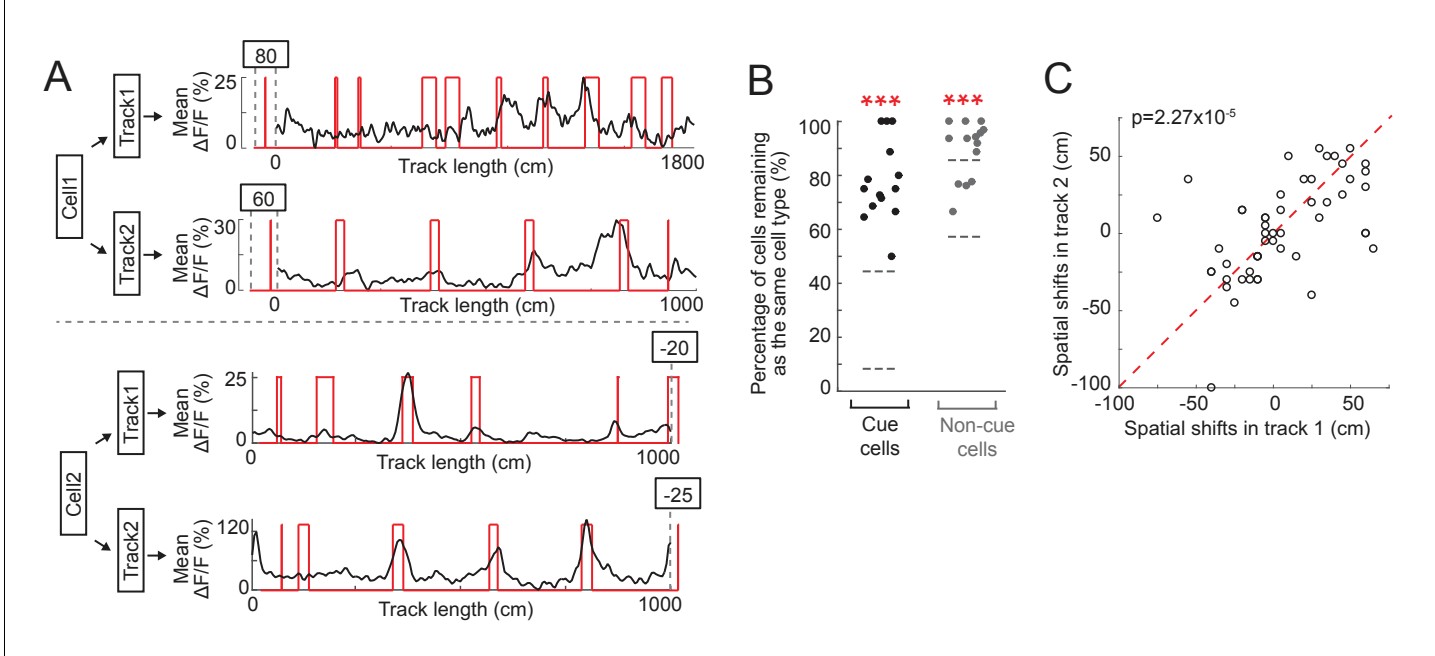

**Figure 6.** Cue cells respond to cues in different environments. (**A**) Examples of two cue cells. Each cell was imaged on two tracks. For each cell: black: mean ΔF/F versus linear track position. Red: cue template. The spatial shift is shown above each plot (right and left shifts of the template relative to the mean ΔF/F curve are defined as negative and positive values, respectively). (**B**) Percentage of cue cells (black) and non-cue cells (gray) that remained as the same cell types in two tracks. Each dot represents one FOV (N = 14 FOVs in 5 mice, which formed 7 groups (two FOVs/group) to determine common cue cells for two tracks). For each cell type, the area between two gray lines represents mean ± STD of data on 50 random datasets, in which cue cells in two tracks were randomly assigned. Real data vs. random data: cue cell: p=1.46 × $10^{-24}$; non-cue cell: p=5.13 × $10^{-14}$. (**C**) Spatial shifts of the same cue cells on two tracks. Each circles show the shifts of one cell. As in A, right and left shifts of the template relative to the mean ΔF/F curve are defined as negative and positive values, respectively. ectively. Red dotted line represents x = y. p=2.27 × $10^{-5}$. ***p≤0.001. Student's t-test.

The online version of this article includes the following source data for figure 6:

**Source data 1.** Percentages of common cue and non-cue cells, and spatial shifts of common cue cells in different environments.

These properties of cue cells suggest that they could provide a source of spatial information in the local circuits of MEC that could be used in error correction in landmark rich environments.

## Cue cells and previously identified cell types

By recording during foraging in real arenas, we were able to determine how cue cells were related to previously identified MEC cell types (grid, head direction, border), all of which are defined by their activity in bounded 2D environments. We found that most cue cells were not grid or border cells, yet they did have noticeably stable, and somewhat irregular, spatial firing patterns in real arenas. Cue cell activity patterns cannot be explained by a relationship to the speed of the animal since animals tended to slow down only at the last cue in the sequence which was associated with a water reward (*Figure 1—figure supplement 2*). Cue cells account for a significant fraction (~22%) of the previously unexplained spatially-stable cells in MEC. While most cue cells were not grid or border cells, a vast majority of them had some orientation tuning (35%). The high prevalence of head direction tuning for these cells suggests either that cue cells may receive inputs from traditional head direction cells, or that a head direction preference is present for cue cells because of the location of particular features of the real arena that drives the activation of cue cells. Further work is required to determine the circuit mechanisms of this orientation tuning preference.

Do cue cells resemble spatially modulated cells in other brain regions, such as place cells or boundary vector cells? Place cells typically have only a single firing field during navigation along short linear tracks, even in virtual reality environments with prominent visual cues along the tracks (*Dombeck et al., 2010*). This is distinctly different from cue cells, in which the number of spatial firing fields scales with the number of cues. Boundary vector cells, which were found in subiculum, encode distance to a boundary (*Lever et al., 2009*; *Stewart et al., 2014*). An identified boundary

vector cell must have a spatial firing field that is uniformly displaced from a particular region of the boundary. The width of the spatial firing field is proportional to the distance from the boundary, meaning that cells shifted significantly from the border would have very wide spatial firing fields, which could cover a large majority of the environment (*Lever et al., 2009*; *Stewart et al., 2014*). Border cells are a special case of boundary vector cells. To determine whether cue cells might be boundary vector cells, we sorted their spatial responses in the real 2D arena based on the shifts of their spatial firing rates from the visual cue pattern on a virtual track (*Figure 3—figure supplement 1*). We found no obvious trends in the spatial firing patterns of these cue cells in the real arena. Along with this, many cells had multiple fields or fields that were not uniformly displaced from the border of the environment. These spatial firing field features of cue cells were inconsistent with those of boundary vector cells. Thus, cue cells have properties distinct from both place cells and boundary vector cells. It is possible that cue cells could have properties similar to landmark vector cells found in the hippocampus (*Deshmukh and Knierim, 2013*). Their spatial firing responses were distributed throughout the arena, despite there being a single white cue card visible, indicating that these cells perform more complex computations in real arenas where spatial cues take many forms in comparison to the cues along a virtual track. Previous studies have also found similar complex responses of non-grid cells in the MEC encoding features of real environments (*Diehl et al., 2017*; *Hardcastle et al., 2017*).

Some recent papers show cell types with features similar to those of cue cells (*Høydal et al., 2019*; *Madisen et al., 2015*; *Wang et al., 2018*). In particular, a recent paper described a new cell type, termed object-vector cells (*Høydal et al., 2019*). We believe these cells belong to a similar cell population as our cue cells do since there are some comparable findings on this cue-related cell type: Similar percentages of cue/object-vector cells; these cells are not predominately conjunctive with other spatial cell types; and these cells maintain cue-related firing across environments (*Figure 6*). In comparison to that paper, our paper provided additional information about these cue/object vector cells. Along with the sequential nature of the activity of these cells, we also specifically studied cue cells in layers 2 and 3 of the MEC and discovered the side-preference of these cells. The fact that these cells mostly responded to cues located on one side and that the right cues were predominately represented in the left MEC, strongly supported a visual input-based mechanism in driving cue cell response.

## Cue cells and path integration

It has been hypothesized that the MEC is a central component of a path integrator that uses self-motion information to update a spatial metric encoded by the population of grid cells (*Burak and Fiete, 2009*; *Fuhs and Touretzky, 2006*; *McNaughton et al., 2006*). Grid cells are grouped into modules based on each cell's grid spacing. Each grid module maintains its own orientation, to which all grid cells align and are related by a two-dimensional spatial phase. Grid cells in a given module maintain their relative spatial phase offsets across different environments (*Fyhn et al., 2007*), including linear tracks (*Yoon et al., 2016*), indicating that the population of grid cells form a consistent spatial metric largely defined only by the two-dimensional phase. This provides support for the idea that grid cell dynamics are constrained to a two-dimensional attractor manifold (*Yoon et al., 2013*). In a manifold-based path integrator, spatial location is represented as the location of the grid cell population activity on the attractor manifold. Self-motion signals, such as running speed in a particular direction, move the population activity along the manifold, such that changes in location are proportional to the integral of the velocity over time. Path integration is inherently a noisy process that requires calibration and error correction for more accurate estimates of position.

In the context of continuous attractor models for path integration, it is interesting to consider the potential functional roles of cue cells. One role could be to act as external error-correction inputs to the path integrator network that tend to drive the neural activity pattern to manifold locations appropriate for each landmark. An analogous use was proposed for border cells, in which they contribute to error correction near boundaries (*Hardcastle et al., 2015*; *Pollock et al., 2018*). If cue cells perform this role independent of other cells, then the cue cell population would need to independently distinguish individual cues as well, one such method would be with cue cells independently varying firing rates at distinct cues. More work is needed to further characterize the nature of this precise coding of unique landmarks and, with new models, to determine how effectively it might

be used to drive an attractor network to the appropriate spatial locations when interacting with a noisy path integrator.

An alternative, or additional, role for cue cells in path integration would be to produce a continuous adjustment of location. The sequence of activity that was produced across the cue cell population as individual landmarks were passed could drive the network activity continuously along the manifold, in essence acting as a velocity input that is quite different from those traditionally considered, such as running speed. This use, as an effective velocity, is analogous to the recent demonstration (*Hopfield, 2015*; *Ocko et al., 2018*) that the collective state of a line attractor can be moved continuously along the manifold by an appropriately learned sequence of external inputs. In essence, the set of inputs at each time point move the location of the activity on the attractor a slight amount, and this is repeated continuously to produce smooth motion, without requiring asymmetric synaptic connectivity and velocity-encoding signals of previous path integrator models (*Burak and Fiete, 2009*; *Fuhs and Touretzky, 2006*; *McNaughton et al., 2006*; *Ocko et al., 2018*). In principle, both path-integration and error correction can be combined through this process.

### Cue cells in real and virtual environments

Why cue cells had such stable and easily classified activity patterns in virtual reality, but not in the real arena remains an open question. Navigating along a virtual track differs greatly from foraging in a complex real arena. In the case of virtual reality, the animal encounters a single cue or pairs of cues at a time, so the representation of location using visual information is primarily from a limited orientation to individual cues or pairs of cues along the track. It is possible that forming a representation of location in the real arena requires triangulation from many cues located at various angles and distances away. Despite the simple design of our real arena with a single cue card on one wall, there could be multimodal features from the floor, walls, or from distal cues outside of the arena. Navigation along simple virtual environments comprised of only visual cues sets up an ideal experimental paradigm to further understand the activity of these cells. Future experiments could probe other features of the cells with more perturbations of the virtual environment.

### Cell classes in MEC

Although our analysis and discussion of cue cells have largely followed the traditional approach of describing MEC cells according to discrete classes, it is interesting to note that cue scores, like grid, head direction, and border scores, each form a continuum and that a significant fraction of cells in MEC are conjunctive for more than one class (*Figure 3*). The conjunctive coding in neural firing in MEC is also demonstrated by a recent study (*Hardcastle et al., 2017*) and is conceptually analogous to the 'mixed selectivity' in neural codes that have been increasingly recognized in cognitive, sensory and motor systems (*Finkelstein et al., 2015*; *Fusi et al., 2016*; *Rigotti et al., 2013*; *Rubin et al., 2014*). Recently, mixed selectivity has been demonstrated to be computationally useful in evidence integration and decision-making by allowing the selection of specific integrating modes in accumulating evidence to guide future behavior (*Mante et al., 2013*; *Ulanovsky and Moss, 2011*). Reframing this in the context of path integration, it will be useful to determine how navigation systems might use mixed selectivity and context-specific integrating modes to weigh different accumulating information (different velocity and position inputs) according to the reliability of that information during navigation in complex, feature-rich environments.

## Materials and methods

**Key resources table**

| Reagent type (species) or resource | Designation | Source or reference | Identifiers | Additional information |
|---|---|---|---|---|
| Strain, strain background (*Adenovirus*) | AAV1.hSyn. GCaMP6f. WPRE.SV40 | Penn Vector Core/addgene | Cat#: 100837-AAV1 | |

*Continued on next page*

*Continued*

| Reagent type (species) or resource | Designation | Source or reference | Identifiers | Additional information |
|---|---|---|---|---|
| Genetic reagent (*M. musculus*) | C57BL/6J | Jackson Laboratory | Stock No: 000664\|Black 6 | |
| Genetic reagent (*M. musculus*) | Thy1-GCaMP6f transgenic line (GP5.3) | Janelia Research Campus; PMID:25250714 | N/A | Male and female |
| Commercial assay | NeuroTrace | Thermo Fisher Scientific | Cat#: N21479 | |
| Software, algorithm | MATLAB | MathWorks | https://www.mathworks.com | |
| Software, algorithm | ImageJ | National Institutes of Health | https://imagej.nih.gov/ij/ | |
| Software, algorithm | ScanImage 5 | Vidrio Technologies | http://scanimage.vidriotechnologies.com/display/SI5/ScanImage+5 | |
| Software, algorithm | ViRMEn (Virtual Reality Mouse Engine) | PMID:25374363 | https://pni.princeton.edu/pni-software-tools/virmen | |
| Software, algorithm | Motion correction (CaImAn) | PMID:30652683 | https://github.com/flatironinstitute/CaImAn | |

## Animals

All procedures were approved by the Princeton University Institutional Animal Care and Use Committee (IACUC protocol# 1910–15) and were in compliance with the Guide for the Care and Use of Laboratory Animals (https://www.nap.edu/openbook.php?record_id=12910). Three C57BL/6J male mice (Stock No: 000664|Black 6), 3–6 months old, were used for electrophysiological experiments. Two 10 week old male mice were used for the two-photon imaging of layer 3 neurons. Mice used for the two-photon imaging of layer 2 neurons were six 10- to 12 week old GP5.3 males and four 10- to 12 week old GP5.3 females, which were heterozygous for carrying the transgene Thy1-GCaMP6f-WPRE to drive the expression of GCaMP6f (*Dana et al., 2014*).

## Experimental design and statistical analysis

All data are represented as mean ± STD. A student's t-test was always used to evaluate whether the difference of two groups of values was statistically significant. Significance was defined using a p value threshold of 0.05 (*$p<0.05$, **$p<0.01$, ***$p<0.001$). All analysis was performed using custom Matlab (MathWorks) software and built in toolkits. All correlations were Pearson correlations unless otherwise specified.

## Real arena for tetrode recording

Experiments were performed as described previously *Domnisoru et al. (2013)*. The real arena consisted of a 0.5 m × 0.5 m square enclosure with black walls at least 30 cm high and a single white cue card on one wall. Animals foraged for small pieces of chocolate (Hershey's milk chocolate) scattered throughout the arena at random times. Trials lasted 10–20 min. On each recording day, real arena experiments were always performed before virtual reality experiments. Video tracking was performed as described previously *Domnisoru et al. (2013)* using a Neuralynx acquisition system (Digital Lynx). Digital timing signals, which were sent and acquired using NI-DAQ cards, and controlled using ViRMEn software in Matlab (*Aronov and Tank, 2014*) were used to synchronize all computers.

## Virtual reality (VR)

The virtual reality system was similar to those described previously (*Dombeck et al., 2010*; *Domnisoru et al., 2013*; *Gauthier and Tank, 2018*; *Gu et al., 2018*; *Harvey et al., 2012*; *Harvey et al., 2009*; *Low et al., 2014*). ViRMEn software (*Aronov and Tank, 2014*) was used to design the linear VR environment, control the projection of the virtual world onto the toroidal screen, deliver water rewards (4 µl) through the control of a solenoid valve, and monitor running velocity of the mice. Upon running to the end of the track, mice were teleported back to the beginning of the track.

### VR for tetrode recording

The animal ran on a cylindrical treadmill, and the rotational velocity of the treadmill, which was proportional to mouse velocity, was measured using sequential sampling of an angular encoder (US Digital) on each ViRMEn iteration (~60 iterations per second). The tracks were 8 meters long with identical cues on both sides of the tracks.

### VR for imaging

Mice ran on an air-supported spherical treadmill, which only rotated in the forward/backward direction. Their heads were held fixed under a two-photon microscope (*Gu et al., 2018*; *Low et al., 2014*). The motion of the ball was measured using an optical motion sensor (ADNS3080; red LED illumination) controlled with an Arduino Due. The VR environment was rendered in blue and projected through a blue filter (Edmund Optics 54–462). The tracks in *Figure 5* and *Figure 5—figure supplement 3* were 10 and 18 meters long, respectively, with asymmetric cues on both sides of the track. Water rewards (4 µl) were delivered at fixed locations of the track (as indicated in the figures). The data in *Figure 6* were collected on three tracks: two tracks with asymmetric cues (10 and 18 meters) and one track with symmetric cues (10 meter). Water rewards (4 µl) were delivered at the beginning and end of the three tracks.

## Microdrives and electrode recording system

Custom microdrives and the electrophysiology recording system used were similar to those described previously *Aronov and Tank (2014)*; *Domnisoru et al. (2013)*; *Kloosterman et al., 2009*. Tetrodes were made of PtIr (18 micron, California Fine Wire) and plated using Platinum Black (Neuralynx) to 100–150 kΩ at 1 kHz. A reference wire (0.004' coated PtIr, 0.002' uncoated 300 µm top) was inserted into the brain medial to the MEC on each side, and a ground screw or wire was implanted near the midline over the cerebellum.

The headstage design was identical to the one used previously *Aronov and Tank (2014)* with the addition of solder pads to power two LEDs for use in tracking animal location and head orientation. Custom electrode interface boards (EIBs) were also designed to fit within miniature custom microdrives. A lightweight 9-wire cable (Omnetics) connected the headstage to an interface board. The cable was long enough (~3 m) to accommodate the moving of the animal between the real arena and the virtual reality system without disconnection.

## Surgery

### Tetrode recording

Surgery was performed using aseptic techniques, similar to those described previously *Domnisoru et al. (2013)*. The headplate and microdrive were implanted in a single surgery that lasted no longer than 3 hr. Bilateral craniotomies were performed with a dental drill at 3.2 mm lateral of the midline and just rostral to the lambdoid suture. After the microdrive implantation, 4–6 turns were slowly made on each drive screw, lowering the tetrodes at least 1 mm into the brain. Animals woke up within ~10 min after the anesthesia was removed and were then able to move around and lift their heads.

### Imaging

The surgical procedures were similar to those described previously *Low et al. (2014)*. A microprism implant was composed of a right angle microprism (1.5 mm side length, BK7 glass, hypotenuse coated with aluminum; Optosigma), a circular coverslip (3.0 mm diameter, #1 thickness, BK7 glass;

Warner Instruments) and a thin metal cylinder (304 stainless steel, 0.8 mm height, 3.0 mm outer diameter, 2.8 mm inner diameter; MicroGroup) bonded together using UV-curing optical adhesive (Norland #81). The microprism implantation was always performed in the left hemisphere (*Gu et al., 2018*; *Low et al., 2014*). A circular craniotomy (3 mm diameter) was centered 3.4 mm lateral to the midline and 0.75 mm posterior to the center of the transverse sinus (at 3.4 mm lateral). The dura over the cerebellum was removed. The microprism assembly was manually implanted, with the prism inserted into the subdural space within the transverse fissure. The implant was bonded to the skull using Vetbond (3M) and Metabond (Parkell). A titanium headplate with a single flange was bonded to the skull on the side opposite to the side of the craniotomy using Metabond. For imaging layer 3 neurons in the MEC, AAV1.hSyn.GCaMP6f.WPRE.SV40 (Penn Vector Core, Cat#: 100837-AAV1) virus was diluted 1:4 in a solution of 20% (w/v) mannitol in PBS and pressure injected at two sites (200 nl/ site): (1) ML 3.00 mm, AP 0.77 mm, depth 1.79 mm; (2) ML 3.36 mm, AP 0.60 mm, depth 1.42 mm.

## Two-photon imaging during virtual navigation

Imaging was performed using a custom-built, VR-compatible two-photon microscope (*Low et al., 2014*) with a rotatable objective. The 920 nm excitation laser was delivered by a mode-locked Ti: sapphire laser (Chameleon Ultra II, Coherent, 140fs pulses at 80 MHz). The laser scanning for imaging layer 2 neurons of the MEC was achieved by a resonant scanning mirror (Cambridge Tech.). The laser scanning for imaging layer 3 neurons of the MEC was achieved by a galvanometer XY scanner (Cambridge Tech.). Fluorescence of GCaMP6f was isolated using a bandpass emission filter (542/50 nm, Semrock) and detected using GaAsP photomultiplier tubes (1077 PA–40, Hamamatsu). The two objectives used for imaging layers 2 and 3 were Olympus 40×, 0.8 NA (water) and Olympus LUCPLFLN 40x, 0.6 NA (air), respectively. Ultrasound transmission gel (Sonigel, refractive index: 1.3359 [*Larson et al., 2011*]; Mettler Electronics) was used as the immersion medium for the water immersion objective used for layer 2 imaging. The optical axes of the microscope objective and microprism were aligned at the beginning of each experiment as described previously *Low et al. (2014)*. Microscope control and image acquisition were performed using ScanImage software (layer 2 imaging: v5; layer 3 imaging: v3.8; Vidrio Technologies *Pologruto et al., 2003*). Images were acquired at 30 Hz at a resolution of 512 × 512 pixels (~410×410 µm FOV) for layer 2 imaging, and 13 Hz at a resolution of 64 × 256 pixels (~100×360 µm FOV) for layer 3 imaging. Imaging and behavioral data were synchronized by simultaneously recording the voltage command signal to the galvanometer together with behavioral data from the VR system at a sampling rate of 1 kHz, using a Digidata/Clampex acquisition system (Molecular Devices).

## Histology

### For tetrode recording

To identify tetrode locations, small lesions were made by passing anodal current (15 µA, 1 s) through one wire on each tetrode. Animals were then given an overdose of Ketamine (200 mg/kg)/Xylazine (20 mg/kg) and perfused transcardially with 4% formaldehyde in 1X PBS. At the end of perfusion, the microdrive/headplate assembly was carefully detached from the animal. The brain was harvested and placed in 4% formaldehyde in 1X PBS for a day and then transferred to 1X PBS. To locate tetrode tracks and lesion sites, the brain was embedded in 4% agarose and sliced in 80 µm thick sagittal sections. Slices were stained with a fluorescent Nissl stain (NeuroTrace, Thermo Fisher Scientific, Cat#: N21479), and images were acquired on an epifluorescence microscope (Leica) and later compared with the mouse brain atlas (Paxinos). To identify which tetrode track belonged to each tetrode of the microdrive, the microdrive/headplate assembly was observed with a microscope to determine the location of each tetrode in the cannula, the relative lengths of the tetrodes, and whether the tetrodes were parallel or twisted. If tetrodes were twisted, then recordings were used only if grid cells were found on the tetrode.

### For imaging

For verifying the layer-specific expression of GCaMP6f in the MEC, animals were transcardially perfused, as described above, and their brains were sliced in 100 µm thick sagittal sections. A fluorescent Nissl stain was performed as described above.

## General data processing for tetrode recording

Data analysis was performed offline using custom Matlab code. Electrophysiology data were first demultiplexed and filtered (500 Hz highpass). Spikes were then detected using a negative threshold set to three times the standard deviation of the signal averaged across electrodes on the same tetrodes. Waveforms were extracted and features were then calculated. These features included the baseline-to-peak amplitudes of the waveforms on each of the tetrode wires as well as the top three principal components calculated from a concatenation of the waveforms from all wires.

### Cluster separation

Features of the waveforms were plotted with a custom Matlab GUI. Criteria for eliminating clusters from the dataset were: units with less than 100 spikes (in real arena or virtual tracks), the minimum spatial firing rate along the virtual track >10 Hz or the maximum firing rate >50 Hz. After this, 2825 clusters remained. Since clusters were cut with two different methods (using all 4 electrodes and using 3 electrodes with the fourth subtracted as a reference), repeats needed to be removed from the overall dataset. Repeats were found using a combination of 3 measures: the Pearson correlation of the real arena spatial firing rate, the same correlation of the virtual track spatial firing rate and the ISI distribution of the spikes merged between the two clusters. If the sum of these scores exceeded 2.25 then the clusters were considered to be from the same cell; the cluster with the larger number of spikes was kept, and the other cluster was discarded.

Recordings were performed on three animals over two months. There were 1081 clusters that were identified on tetrodes that were histologically identified to be in MEC during the recording that also passed our cluster quality criteria. The grid scores of these cells were calculated and a grid score threshold was calculated using shuffled permutations of these cells (*Aronov and Tank, 2014*; *Domnisoru et al., 2013*). Any tetrode on a particular day with a grid cell was then added to the database from that date on (2107 clusters). Duplicate units that were recorded across multiple days were removed using the correlation of firing rates with mean subtraction performed of both their real arena and virtual track spatial firing rates. The cell with the larger maximum spatial firing rate in the arena was kept if the correlation between spatial firing rates was $\geq 0.8$ or if the correlation between virtual track rates was $\geq 0.75$. There were 789 clusters remaining after duplicates were removed.

## Spatial firing rates of tetrode data

Position data (including head orientation in real 2D arenas) were subsampled at 50 Hz and spikes were assigned into the corresponding 0.02 s bins. Velocity was calculated by smoothing the instantaneous velocity with a moving boxcar window of 1 s. Only data in which the animal's smoothed velocity exceeded 1 cm/sec were used for further analyses of firing rates or scores.

### Real arena spatial firing rate

2D arenas were divided into 2.5 × 2.5 cm bins. Spike counts and the total amount of times spent in these bins were convolved with a Gaussian window (5 × 5 bins, σ = 1 bin). Firing rate was not defined for spatial bins visited for a total of less than 0.3 s.

### Real arena, head direction

The animal's head direction was binned in 3-degree intervals. For each angle bin, the spike count and the total amount of time spent (occupancy) was calculated. These values were separately smoothed with a 15 degree (5 bins) boxcar window, and the firing rate was computed as the ratio of the smoothed spike count to the smoothed occupancy.

### Virtual track spatial firing rate

Virtual tracks were divided into 5 cm bins. Spike counts and the amount of time spent in these bins were smoothed independently with a Gaussian window (3 point, σ = 1). The smoothed firing rate was calculated as the smoothed spike position distribution divided by the smoothed overall position distribution.

## Spatial firing fields

To calculate spatial firing fields we created time arrays for position and for number of spikes of a cell. Time bins were 100 msec. For position, we calculated the average position within each chunk of time (5 data points since data were interpolated to 20 msec sampling intervals). We then divided the spatial track into 5 cm bins and determined in which bin the average position was located. For spikes, we counted the number of spikes in that 100 msec interval. This generated two arrays in time (sampled at 100 msec), one with spike count and one with spatial bin location along the track. We then circularly permuted the spike count array by a random time interval between 0.05 x recording length and 0.95 x recording length. We then calculated the smoothed firing rate of this shuffled spike time array with the spatial bin location array. This was repeated 100 times, and the shuffled spatial firing rate was calculated for each permutation. The p-value was defined for each spatial bin along the track as the fraction of permutations on which the firing rate in that bin was above the actual firing rate. Any bin in which the p-value was less than 0.3 was considered part of a firing field.

## Firing field distributions

For each cue cell, we defined an array (5 cm bins) that is 1 when there is a firing field and 0 otherwise. To look at the distribution of firing fields for the population of cells, we summed the values for each bin across all cue cells and divided by the number of cells. This gave the fraction of cue cells with firing fields at each location. The plot of this fraction versus location was defined as the population firing field distribution.

## Scores for cells in tetrode data

For the cue score, spatial firing rates remained the same and the cue score was shuffled with cues randomly redistributed along the track. For ridge/background ratio, spatial field shuffles were performed. For all other scores, the shuffle was performed with spike times circularly permuted by a random amount of time chosen between 0.5 x recording length and 0.95 x recording length, a standard method for determining score thresholds (*Domnisoru et al., 2013*). Shuffled distributions from all units combined were used to calculate a threshold at 95th percentile.

## Grid score

The unbiased autocorrelation of the 2D firing rate in a real arena was first calculated (*Hafting et al., 2005*). Starting from the center of the 2D autocorrelation function, an inner radius was defined as the smallest radius of three different values: local minimum of the radial autocorrelation; where the autocorrelation was negative; or at 10 cm. Multiple outer radii were used from the inner radius + 4 bins to the size of the autocorrelation - 4 bins in steps of 1 bin. For each of these outer radii, an annulus was defined between the inner and the outer annulus. We then computed the Pearson correlation between each of these annuli and its rotation in 30 degree intervals from 30 to 150 degrees. For each annulus we then calculated the difference between the maximum of all 60 and 120 rotation correlations and the minimum of all 30, 90, and 150 degree correlations. The grid score was defined to be the maximum of all of these values across all annuli. 100 shuffles for each cell were performed and pooled.

## Head direction score

The head direction score was defined to be the mean vector length of the head direction firing rate (*Giocomo et al., 2014*). The head direction angle was defined to be the orientation of the mean vector of the head direction firing rate. 100 shuffles for each cell were performed and pooled.

## Border score

Border scores were calculated as described in the original publication describing this cell type (*Solstad et al., 2008*). 100 shuffles for each cell were performed and pooled.

## Spatial/head direction stability

This was calculated as described previously *Boccara et al. (2010)*. Recording sessions were divided into two parts, the firing rate was calculated for each, and the spatial stability was defined as the Pearson correlation between the two parts. 100 shuffles for each cell were performed and pooled.

## Cue score

The cue score was developed to measure the correlation of the spatial firing rate to the visual cues of the environment. A 'cue template' was defined in 5 cm bins with value equal to 1 for bins that included the area between the front and back edges of each cue and 0 elsewhere. The cross correlation between the cue template and the firing rate was first calculated (relative shift ≤300 cm). The peak in the cross correlation with the smallest absolute shift from zero was chosen as the best correlation of the firing rate to the cue template. The spatial shift at which this peak occurred was then used to displace the cue template to best align with the firing rate. The correlation was then calculated locally for each cue. The local window included the cue and regions on either side extending by half of the cue width. The mean of local correlation values across all cues was calculated and defined as the 'cue score'. An illustration of this method is shown in *Figure 1B*. This score effectively distinguished grid cells from cue cells, because grid cells generally did not have peaks at consistent locations relative to all the cues. The small number of grid cells that passed the cue score shuffle test also tended to have activity in other locations, where cues were not present. 100 shuffles for each cell were performed and pooled.

## Ridge/background ratio

The ridge/background ratio was calculated on the smoothed spatial firing rate at each cue location. The spatial firing rate of each cell was shifted to maximally align to the cue template as was done to calculate the cue score. The 5 bins (25 cm) in the center of each cue location are defined to be bins for the ridge. Background bins were all bins outside of cue locations displaced in both directions by [cue half-width + 20] to [cue half-width + 30]. For each cue, the ridge/background ratio was calculated as the mean firing rate in the ridge bins divided by the mean firing rate in the background bins. The ridge/background ratio for the cell was defined to be the mean of these individual ridge/background ratios. We performed 1000 shuffles of the data, as described above, and calculated the mean ridge/background ratio for each shuffle. The p value is the (number of shuffled data mean values larger than the mean ridge/background ratio of the data)/(number of shuffled data mean values less than the mean ridge/background ratio of the data).

## General imaging data processing

All imaging data were motion corrected using a whole-frame, cross-correlation-based method (*Dombeck et al., 2010*) and were then used to identify regions of interest (ROIs) with fluorescence changes occurring during virtual navigation using an independent component analysis (ICA) based algorithm (*Mukamel et al., 2009*) (for individual layer 3 field of view (FOV): mu = 1, 150 principal components, 150 independent components, s.d. threshold = 3; for individual layer 2 FOV, which was evenly split as nine blocks before ICA: mu = 0.7, 30 principal components, 150 independent components, s.d. threshold = 3). Fluorescence time series of these ROIs were extracted from all motion-corrected stacks. The fractional change in fluorescence with respect to baseline (ΔF/F) was calculated as $(F(t) - F_0(t)) / F_0(t)$, similar to what was described previously *Gu et al. (2018)*; *Low et al. (2014)*. Significant calcium transients were identified as those that exceeded cell-specific amplitude/duration thresholds (so that artefactual fluctuations were expected to account for less than 1% of detected transients *Dombeck et al., 2007*). Mean ΔF/F of the whole imaging session or individual traversals was calculated as a function of position along the virtual track for non-overlapping 5 cm bins. Only data points during which the mouse's running speed met or exceeded 1 cm/s were used for the calculation.

## Identifying cue cells in imaging data

### Selection of cells

candidates for cue cells were restricted to cells that contained at least one in-field period and one out-of-field period based on a p-value analysis of their calcium responses (*Domnisoru et al., 2013*; *Gu et al., 2018*; *Heys et al., 2014*; *Yoon et al., 2016*). Similar to identifying spatial fields for tetrode-recorded cells, in- and out-of-field periods were defined by comparing the mean ΔF/F value in each 5 cm bin to that of a random distribution created by 1000 bootstrapped shuffled responses, which were generated by rotating the ΔF/F trace starting from random sample numbers between $0.05 \times N_{samples}$ and $0.95 \times N_{samples}$ ($N_{samples}$: number of samples in the ΔF/F trace). For each bin,

the p-value equaled the percent of shuffled mean ΔF/F that were above the real mean ΔF/F. In-field-periods were defined as three or more adjacent bins (except at the beginning and end of the track where two adjacent bins were sufficient) whose p-value$\leq$0.2 and for which at least 10% of the runs contained significant calcium transients within the period. Out-of-field periods were defined as two or more bins whose p-value$\geq$0.75.

## Calculating cue scores to left and right cue templates and defining left and right cue cells

Left and right cue templates were generated using the locations of cues on the left and right sides of the track, respectively. Left and right cue scores for each cell to the left and right templates were calculated as described above (Scores for cells in tetrode data, *Cue score*). Cue cells were defined as those with cue scores above the threshold, which was the 95$^{th}$ percentile of shuffle scores of all cells. For each cue cell, 200 shuffle scores were computed as its cue scores on 200 shuffled templates, which contained cues identical to the original template but arranged at random locations of the track.

Using the above method, we assigned the cells that uniquely passed left and right template thresholds as right and left cue cells, respectively. Moreover, since the bimodal distribution of bilateral scores (explained below) of the left and right cue cell populations (*Figure 5E*) indicated their primary responses to single-side cues, we assigned cells that passed the thresholds of both left and right templates to the side with higher cue scores.

### Bilateral scores

Bilateral score was defined as the difference between the left and right cue scores (left cue score – right cue score) when a cell response was aligned to its preferred template (*Figure 5—figure supplement 2B*). For a right cue cell with right cue score R1 under a spatial shift S1, its left cue score L1 was calculated when the cell's response was aligned to the left cue template under the same spatial shift S1. Its bilateral score = L1 – R1. For a left cue cell with left cue score L2 under a spatial shift S2, its right cue score R2 was calculated when the cell's response was aligned to the right cue template under the same spatial shift S2. Its bilateral score = L2 – R2. The bilateral score of a cell preferentially responding to left or right side cues should have a large absolute value, whereas the bilateral score of a cell equally responding to left and right side cues should have a small absolute value near zero.

## Acknowledgements

We thank current and former members of the Tank lab, Ila Fiete, and Anika Kinkhabwala for helpful discussions, and Jeffrey Santner and Alexander Riordan for comments on the manuscript. This work was supported by NINDS Grant 5R37NS081242 (DWT), NIMH Grant 5R01MH083686 (DWT), NIH Postdoctoral Fellowship Grant F32NS070514-01A1 (AAK).

## Additional information

### Funding

| Funder | Grant reference number | Author |
| --- | --- | --- |
| National Institute of Neurological Disorders and Stroke | 5R37NS081242 | David W Tank |
| National Institute of Mental Health | 5R01MH083686 | David W Tank |
| National Institutes of Health | F32NS070514-01A1 | Amina A Kinkhabwala |

The funders had no role in the experiments or analysis in this publication.

### Author contributions

Amina A Kinkhabwala, Conceptualization, Resources, Data curation, Software, Formal analysis, Funding acquisition, Validation, Investigation, Visualization, Methodology; Yi Gu, Data curation, Formal

analysis, Investigation, Visualization; Dmitriy Aronov, Conceptualization, Data curation, Software, Formal analysis, Methodology; David W Tank, Conceptualization, Resources, Data curation, Software, Formal analysis, Supervision, Funding acquisition, Validation, Investigation, Methodology

### Author ORCIDs
Amina A Kinkhabwala (ID) https://orcid.org/0000-0002-0778-1677
David W Tank (ID) https://orcid.org/0000-0002-9423-4267

### Ethics
Animal experimentation: All procedures were approved by the Princeton University Institutional Animal Care and Use Committee (IACUC protocol# 1910-15) and were in compliance with the Guide for the Care and Use of Laboratory Animals.

### Decision letter and Author response
Decision letter https://doi.org/10.7554/eLife.43140.sa1
Author response https://doi.org/10.7554/eLife.43140.sa2

## Additional files

### Supplementary files
• Transparent reporting form

### Data availability
All data generated or analyzed during this study are included in the manuscript and supporting files.

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
