## [Decision Letter]

**Acceptance summary:**

It is important to understand how path integration errors are corrected in MEC. Your demonstration of cue cells in MEC suggests an interesting mechanism complementing border cells in performing this task.

**Decision letter after peer review:**

Thank you for submitting your article "Visual cue-related activity of cells in the medial entorhinal cortex during navigation in virtual reality" for consideration by *eLife*. Your article has been reviewed by three peer reviewers, one of whom is a member of our Board of Reviewing Editors, and the evaluation has been overseen by Laura Colgin as the Senior Editor. The reviewers have opted to remain anonymous.

The reviewers have discussed the reviews with one another and the Reviewing Editor has drafted this decision to help you prepare a revised submission.

Summary:

The authors use a combination of tetrode recording and calcium imaging while mice ran on virtual linear tracks containing visual cues (towers) on either side to describe a novel cell type in the superficial layers of medial entorhinal cortex (MEC), which they call "cue cells." Correlation based measures using the spatial firing rate of the cells and a landmark cue template were used to classify cells as "cue cells" that respond by firing repeatedly at every landmark. Recordings in two-dimensional open-field environments revealed the cue cells were also conjunctively encoding other, previously characterized, features of cells in the mEC including the presence of borders (border cells), firing in a regular triangular spatial lattice pattern (grid cells) and the animals heading direction (head direction cells, ~50% of cue cells had some orientation tuning). The results were viewed by all reviewers as novel and significant.

However, there are multiple significant concerns, as the manuscript stands, listed below, which the authors should be able to address in the allotted time. The major issues concern the shuffling procedure used to generate control distributions for cue scores, missing details regarding statistics, and other analyses that need to be modified to better control for spatially selective cells that are not anchored to cues.

Essential revisions:

1) The circular permutation procedure undertaken to form a control data set is inadequate for many of the analyses it is used in. Circular permutation destroys spatial selectivity. When the question being addressed is whether the observed spatial selectivity is correlated with the landmark locations (rather than being random), the control distribution of correlation coefficients needs to be obtained using a randomization procedure maintaining spatially selectivity while randomizing the relative positions of the place fields and the objects. A control distribution of correlation coefficients without spatial selectivity, like the one used here, is likely to be more tightly clustered around zero than a control distribution of correlation coefficients obtained from spatially selective (but randomized) data, thus allowing more false positives. One easy way to do this is to randomize landmark locations while keeping the ratemap unchanged. This will work for detection of cue cells, but for some analyses (e.g. the sequence analysis), shuffling the place fields within a ratemap would be a more suitable shuffle.

2) Some of the findings appear to follow tautologically from the definition of the cue score, which correlates firing patterns to a template that matches the locations of the visual cues. a) The distribution of this score appears uni-modal and the authors them pick out one end of the distribution. Are the cue cells really then a discrete population? Following this, are the cue locations really special, or does the template just pick out cells that fire near the cues from amongst a population that uniformly spans the environment? Can you compare the number of cells you would identify with a randomized cue template to the number of cells picked up by the cue template? The cue-score method picks out cells with positive correlations to the cue template. Are there cells with significant negative correlations? ("anti-cue cells")

b) Do cells that fire near the visual cues respond more to the removal of visual cues than cells that fire away from the visual cues, or do all cells lose their spatial tuning in the cue-removed condition?

c) Can the fact that the sequence is repeated at each cue be explained by the fact that the cue template looks for cells with a fixed spatial offset from each cue? One way to control for this would be to identify cells based on one cue only, and test whether the sequence repeats at the other cues. Alternatively, a cue score could be developed that allows the cue template to move independently at each cue. This would be a more convincing test that cells really do have a fixed offset from each spatial cue.

d) What is the distribution of Left-Cue scores for Right-Cue cells and vice versa? Is it really "either/or", or is there a continuum of cells that respond to combinations of left and right cues?

e) "For each environment, we found the activity of all cue cells was best aligned to the center of the cue rather than the start or end of the cues". If the place field size is proportional to the cue size, this will automatically follow. Start and end would be displaced differentially with respect to place field center while the cue center would be the average of the two. The method used for generating cue scores (subsection “Scores for cells in tetrode data”) generates higher scores for cells with field sizes matching cue sizes (over cue cells that have cue independent field sizes), making this analysis circular. Why not use the peak correlation between the cue template and the firing rate map with the smallest absolute shift from zero as the cue score, instead? That will eliminate this confound.

3) Detailed statistics need to be provided at multiple places. For example, the subsection “Cue cell pairwise activity patterns” mentions Pearson correlation coefficients of 0.3 and 0.13. The authors argue that "This suggests that the spike timing relationship between cue cell pairs is present only when cues are present and thus when these cells are driven to be active in a sequential manner by locomotion past the cue." This will hold true if the two coefficients are significantly different from one another, and the coefficient of 0.3 is statistically significantly different from 0. 2. "*p ≤0.05. **p ≤0.01. ***p ≤0.001. n.s. p > 0.5. Student's t-test. Error bars: mean ± SEM.": detailed statistics including sample size, p values, t-statistics, means and STDs need to be reported in the main text. The journal guidelines state "Report exact p-values wherever possible alongside the summary statistics and 95% confidence intervals. These should be reported for all key questions and not only when the p-value is less than 0.05."

Have the authors corrected for multiple comparisons wherever required?

4) The authors report recording up to 301 cells from a single tetrode (Figure 1—figure supplement 1; including 88 cue cells and 93 grid cells; "Recordings were performed on four animals over two months."). Were repeat recordings from the same neurons on consecutive/multiple days identified and eliminated? How? If they were not eliminated, all the reported statistics suffer from inflation of degrees of freedom. Can the authors comment on this?

5) What are the running speed profiles of the mice? Did they tend to slow down near the visual cues?

6) "In layers 2 and 3, we consistently observed that anatomically adjacent cue cells (physical distances around 30 μm) showed more similar spatial shifts, whereas the relationship was more varied if cue cells were further apart (Figure 7G-N). The similar cue responses of adjacent cue cells suggest that they may share similar inputs or be connected."

Do the cross correlations of neighbouring cells (on the same tetrode) maintain the peak at 0ms in B if they had peak at 0ms in A? If not, the two observations with tetrodes and imaging would contradict one another. In general, the claims made in G-N are rather weak. Authors should consider excluding them.

The 'micro-organisation' relating physical separation to spatial shifts in responses relative to cue location is seen restricted to anatomically adjacent cue cells: is there any danger that this reflects contamination/poor localisation/diffusion of light from neighboring sources?

7) One reason for the more specific apparent correlate of firing in the VR track versus the open field might be that the viewing angle is important and this is not systematically sampled in the open field. Do the mice run in both directions on the VR – if so, do the cue cells fire at a similar cue-angle? Does this relate to the observation that place cell firing becomes more directionally modulated in VR than in real open fields (presumably because of the greater influence of vision in visually-generated VR; Acharya, Aghajan et al., 2016; Chen, King et al., 2018). In the comparison to known cell types, might these cue cells be related to landmark-vector cells (Deshmukh and Knierim, 2013) or object-vector cells (Hoydal et al., 2019), or egocentric responses recently reported in lEC (Wang et al., 2018)? Do the cue cell responses depend on the wider context – need the mouse be running (vs. passive viewing) to see firing? To what extent do cue cells fire similarly across VR environments or do they 'remap'? (this is not entirely clear from Figure 1).

[Editors' note: further revisions were suggested prior to acceptance, as described below.]

Thank you for submitting your article "Visual cue-related activity of cells in the medial entorhinal cortex during navigation in virtual reality" for consideration by *eLife*. Your article has been reviewed by two peer reviewers, one of whom is a member of our Board of Laura Colgin as the Senior Editor. The reviewers have opted to remain anonymous.

The reviewers have discussed the reviews with one another and the Reviewing Editor has drafted this decision to help you prepare a revised submission.

The reviewers remain positive about the overall importance of these results.

However, although some of the concerns were adequately addressed during the first revision, some major concerns that were raised in the first round of review remain. Moreover, a number of new concerns arose from the revisions. Still, reviewers are confident that the authors can easily address these remaining concerns.

Essential revisions:

Duplicates removed database in Author response image 7, which is used throughout the paper uses spatial firing rate correlations > 0.95 as a threshold for discarding cells as being duplicates. This threshold is unreasonably high, as even stable place cells recorded in consecutive sessions in the hippocampus often have substantially lower correlation coefficients, especially in mice (e.g. Kentros et al., 1998, Figure 3). This means that the duplicates removed database is likely to still have an unreasonably high number of duplicates.

The authors should show the tetrode lowering database figure shown in Author response image 7 at least as supplementary data. They must also include the tetrode lowering database stats and aggregate figures for other analyses using tetrode data, including responses to environmental perturbations, sequences, pairwise correlations etc. to convince the reader that the significance of the patterns reported is not grossly overestimated by inflation of degrees of freedom caused by the inclusion of duplicates in their dataset.

In the revised manuscript, it is no longer clear how many animals the data were collected from, and how many neurons of different types were contributed by each animal. The animal and tetrode – wise breakdown of neurons in the tables included in the previous version are essential. Tables showing number of units of different kinds recorded from each tetrode in each animal have been eliminated from Figure 1—figure supplement 1. They should be put back, with numbers for both duplicates removed database as well as for tetrode lowering database.

Related to this, it is not clear how many animals contributed to the new data shown in the new Figure 7. Hence, it is impossible to figure out if the reported results are reproducible across animals. Please mention number of animals included for each analysis/figure in the Results.

"In Region A, there was a spread in the temporal shifts for pairs of cue cells and these temporal shifts were correlated for the two tracks (Figure 5C left, Pearson correlation = 0.52, p=9×10-5). However, the temporal shifts in Region B of the two tracks were less correlated: while a similar spread of temporal shifts was observed when cue cells were recorded on the with cues track (plotted along the x-axis of the bottom right panel in Figure 5C), most cue cell pairs did not have a correlated phase in the relative spike timing when cues were missing (plotted along the y-axis of the bottom right panel in Figure 5C right, “correlation not significant”). The fact that the spike timing relationship between cue cell pairs is maintained only when cues are present suggests that these cells are driven to be active in a sequential manner by locomotion past cues."

Correlation coefficient and p value for region B needs to be included, as requested in the previous review – stating "correlation not significant" is not sufficient. Furthermore, to make the claim that their data suggests that "cells are driven to be active in a sequential manner by locomotion past cues", the authors should demonstrate that the slopes in region A and B shown in Figure 5C are significantly different from one another.

---

## [Author Response]

Essential revisions:1) The circular permutation procedure undertaken to form a control data set is inadequate for many of the analyses it is used in. Circular permutation destroys spatial selectivity. When the question being addressed is whether the observed spatial selectivity is correlated with the landmark locations (rather than being random), the control distribution of correlation coefficients needs to be obtained using a randomization procedure maintaining spatially selectivity while randomizing the relative positions of the place fields and the objects. A control distribution of correlation coefficients without spatial selectivity, like the one used here, is likely to be more tightly clustered around zero than a control distribution of correlation coefficients obtained from spatially selective (but randomized) data, thus allowing more false positives. One easy way to do this is to randomize landmark locations while keeping the ratemap unchanged. This will work for detection of cue cells, but for some analyses (e.g. the sequence analysis), shuffling the place fields within a ratemap would be a more suitable shuffle.

We have changed our shuffling methods based upon this comment. All major results remain the same in both tetrode and imaging data.

1) For sequence analysis in Figure 4 of the manuscript, shuffling of spatial fields was used for the ridge/background calculations as suggested.

2) To calculate a cue score threshold for identifying cue cells, instead of circularly permuting spike times or fluorescence signals to calculate shuffled distributions, we tried both methods suggested above, shuffling of cues on the template and spatial fields of the spatial firing rate. We have plotted the shuffle distributions in Author response image 1 since there was concern about the shape and clustering of the shuffle distributions. We found no significant differences in the distributions and threshold values for any of the three methods (Author response image 1). Since shuffling cues on the template generally gave higher thresholds, we used this method to identify cue cells. All major conclusions remained the same.

3) In addition to the threshold changes, for imaging data, we also introduced template-specific thresholds to address Essential Revision 2d. Only the data of left and right cue cells are included in the current manuscript and the details are explained in Essential Revision 2d.

**Author response image 1. respfig1:** Score thresholds calculated by different shuffling methods. (A) The distribution of shuffled cue scores and thresholds using three shuffling methods: circular permutation (circular shuffle), spatial field shuffle (field shuffle), and cue template shuffle (template shuffle). The thresholds values (95^th^ percentile of shuffles) are indicated by dark blue, red and light blue lines, respectively. Left: distribution of real and shuffled cue scores for tetrode and imaging data. Right: distribution of real and shuffled cue scores for imaging data. Thresholds for left, right and both-side templates were separately calculated. B) Comparison of the thresholds generated by all three shuffling methods for tetrode and imaging data. In the current manuscript, we used the template shuffling method (highlighted in yellow).

2) Some of the findings appear to follow tautologically from the definition of the cue score, which correlates firing patterns to a template that matches the locations of the visual cues. a) The distribution of this score appears uni-modal and the authors them pick out one end of the distribution. Are the cue cells really then a discrete population? Following this, are the cue locations really special, or does the template just pick out cells that fire near the cues from amongst a population that uniformly spans the environment? Can you compare the number of cells you would identify with a randomized cue template to the number of cells picked up by the cue template? The cue-score method picks out cells with positive correlations to the cue template. Are there cells with significant negative correlations? ("anti-cue cells")

If we gave the impression that we think they are a discrete population, we want to clarify that they are not. We have gone through the paper to make sure our wording is consistent with this. As described in a recent paper from the Giocomo lab, we agree that the MEC is comprised of cell types with some degree of mixed selectivity (Hardcastle et al., 2017).

Based on the reviewer’s suggestion, we further investigated whether there is a preferred representation of cues of the environment by comparing the percentage of identified cue cells using the cue templates of the current environment to those identified using templates of random environments. As shown in Figure 5—figure supplement 3, we found that the percentage of cue cells identified in the current environment was significantly higher than that in random environments. This indicates that cue cells are a unique population within MEC rather than a subpopulation picked out from a larger population of cells with spatial fields distributed across the track. This result also shows that the environmental cues are preferentially represented by cue cells

b) Do cells that fire near the visual cues respond more to the removal of visual cues than cells that fire away from the visual cues, or do all cells lose their spatial tuning in the cue-removed condition?

We compared the spatial firing patterns of cue cells with fields near cues (within a 25 cm spatial shift between the spatial firing rate and cue template) to those with fields far from cues (greater than 25 cm spatial shift). In Author response image 2, the spatial firing fields for all cells with fields near and far from cues are shown on the left (Author response image 2A1) and right (Author response image 2A2), respectively. In each case, the top plots show fields of all cells and the population distribution (cell fraction with fields at each 5 cm bin along the track) for the track in which all cues are present. The bottom two panels show the fields for the track with cues missing along the latter part of the track. In both cases, fields are present where cues are present, and are missing in places where cues have been removed. In Author response image 2, the fraction of 5 cm bins that have a spatial field (field bins) is plotted for these two conditions (fields near cues and far from cues). For cue cells with fields near and far from cues, there is a decrease in the fraction of field bins when cues are removed. Both of these results are statistically significant, but with different p values: paired one-tailed t-tests: for cue cells with fields near cues in Region B: with cues field bin fraction > missing cues field bin fraction, N = 77, p = 8×10^-11^; for cue cells with fields far from cues in Region B: with cues field bin fraction > missing cue field bin fraction, N = 20, p = 4×10^-5^. There was no significant difference in the responses in Region A for either category of cells. Therefore, the spatial firing patterns of cue cells with fields near cues and far from cues are both similarly affected by the removal of cues.

**Author response image 2. respfig2:** Cue-removal responses of cue cells with fields near or far from cues. (A) Cue cells were sorted into two groups based on the spatial shift of their spatial firing rate relative to the cue template. Cells with spatial shifts less and more than 25 cm in either direction were categorized as cells with fields near and far from cues, respectively. a1: Top: spatial firing fields of all cue cells with fields near cues and the fraction of cells within each 5 cm bin with firing fields. Bottom: spatial firing fields and distribution of firing fields for the population of cells with fields near cues is shown for the track with the cues missing along the latter part of the track. a2: Similar to the plots in a1 but for cells with spatial firing fields far from cues. B) Summary plots: the fraction of bins with fields (field bins) for each cell is plotted for cells with fields near and far from cues on different track regions. b1: summary plots for cells with fields near cues. Left: the fraction of field bins for each cell in the early region of the track in which cues are present for both the “With cues” and “Missing cues” tracks (Region A). Right: the same type of plot for the latter part of the track where cues are present on the “With cues” track and missing on the “Missing cues” track. b2: summary plots for cells with fields far from cues. The dotted lines are drawn diagonally (slope=1).

c) Can the fact that the sequence is repeated at each cue be explained by the fact that the cue template looks for cells with a fixed spatial offset from each cue? One way to control for this would be to identify cells based on one cue only, and test whether the sequence repeats at the other cues. Alternatively, a cue score could be developed that allows the cue template to move independently at each cue. This would be a more convincing test that cells really do have a fixed offset from each spatial cue.

We agree with the reviewer’s concern that cells with consistent spatial shifts from individual cues could be artificially selected based on the classification criteria of cue cells. To avoid this artifact, the ideal analysis would be to classify cells using a single cue and measuring their spatial shifts to other cues, as the reviewer suggested. However, we found that a template comprised of only one cue picked up a large number of other cell types, including grid cells, the activity of which did not correlate to other cues along the track. Based on our observations of the data collected on the current tracks, three to four cues on a track are generally required to specifically select cells with cue-correlated activity.

We performed a modified version of the suggested analysis on calcium imaging data since the environments used in the previous version of the paper were comprised of more cues (eight) than the tetrode data environments (Figure 5—figure supplement 5). In general, for a given cue template, we classified cells with cue-correlated activity using a half-template containing five cues (template 1), and then compared its spatial shift on template 1 to that of the other half-template comprised of the rest five cues (template 2). The hypothesis is that the spatial shift will be similar for these two half-templates if cue cell responses are similarly shifted from all cues. An example with two half-templates is shown in Figure 5—figure supplement 5A. R1 and R2 are two half-templates with cues on the right side of the track. We calculated the percentage of cells that maintained similar spatial shifts across the two half-templates (where the difference of the spatial shifts on R1 and R2 is less than 25 cm). We found that a large fraction of cue cells (76.9% and 80.3% for cells identified on R1 and R2, respectively) had very similar shifts on the two half-templates.

To further confirm that this high percentage of cells with consistent spatial shifts was not created by using a particular set of half-templates, we repeated this analysis for cells in both layers 2 and 3 using multiple sets of half-templates comprised of various combinations of cues from the original templates. We also performed the analysis for left-side cues (Figure 5—figure supplement 5B). All analyses showed similar results (Figure 5—figure supplement 5C). These data together indicate that responding to individual cues at consistent spatial shifts is a feature of most cells with cue-correlated activity. We can include this result in a supplementary figure if necessary.

d) What is the distribution of Left-Cue scores for Right-Cue cells and vice versa? Is it really "either/or", or is there a continuum of cells that respond to combinations of left and right cues?

In Figure 5—figure supplement 2A, we plotted the distributions of left and right cue scores of cells in the new environment and found that they both exhibited unimodal, continuous distributions. Most cells had cue scores above the cue score threshold for left or right cues but not both. Only a small fraction (~5% among all cells classified using left and right cue template and ~1.6% of all cells) of cells passed the thresholds of both left and right cue templates (top right corner). As shown in Figure 6—figure supplement 2A, their responses correlated to the two templates under different spatial shifts, indicating that they did not simultaneously responded to both left and right cues. Furthermore, we developed a “bilateral score” to specifically determine whether a cell response was encoding cues on one side or both-sides of the environment (Figure 6—figure supplement 2B). Bilateral scores of the left and right cue cells together showed a bimodal distribution (Figure 6E), suggesting that these cells responded to either left or right cues, but not both.

To address conjunctive left-cue/right-cue cells more directly, we also previously used a template with cues on both sides. We classified cells using a threshold specific to the both-side template of the new environment (different from the method in the previous version of the manuscript, where we used a single threshold for all three types of template, Figure 5—figure supplement 2C). However, from closer inspection, we chose to remove the bothside cue cells from the main paper for the following reasons:

1) The cue scores of both-side cue cells were significantly lower than those of left and right cue cells (Figure 5—figure supplement 2D). Since cue score was the mean correlation of a cell response to individual cues, independent of the number of cues on a template, the low cue scores indicated that the responses of both-side cue cells did not correlate well to cues on both-sides of the track.

2) 64% of both-side cue cells were also classified as left or right cue cells, which only strongly responded to cues on one side (see the sequence plots in Figure 5—figure supplement 2G).

3) The rest of both-side cue cells (36%) only weakly correlated to the both-side template (Figure 5—figure supplement 2E-G), as reflected by their lower cue scores (Figure 5—figure supplement 2D).

These conclusions were consistently obtained in cells in layers 2 and 3 of the MEC imaged in the previous environment (Figure 6—figure supplement 3 and Figure 5—figure supplement 4C-D and H-I). Since we believe that cue cells preferentially respond to cues on either left or right cues but not both, we chose to focus on left and right cue cells in the current manuscript.

e) "For each environment, we found the activity of all cue cells was best aligned to the center of the cue rather than the start or end of the cues". If the place field size is proportional to the cue size, this will automatically follow. Start and end would be displaced differentially with respect to place field center while the cue center would be the average of the two. The method used for generating cue scores (subsection “Scores for cells in tetrode data”) generates higher scores for cells with field sizes matching cue sizes (over cue cells that have cue independent field sizes), making this analysis circular. Why not use the peak correlation between the cue template and the firing rate map with the smallest absolute shift from zero as the cue score, instead? That will eliminate this confound.

We agree with the reviewer and have removed Figure 4C-F and the sentence that was quoted.

3) Detailed statistics need to be provided at multiple places. For example, the subsection “Cue cell pairwise activity patterns” mentions Pearson correlation coefficients of 0.3 and 0.13. The authors argue that "This suggests that the spike timing relationship between cue cell pairs is present only when cues are present and thus when these cells are driven to be active in a sequential manner by locomotion past the cue." This will hold true if the two coefficients are significantly different from one another, and the coefficient of 0.3 is statistically significantly different from 0. 2. "*p ≤0.05. **p ≤0.01. ***p ≤0.001. n.s. p > 0.5. Student's t-test. Error bars: mean ± SEM.": detailed statistics including sample size, p values, t-statistics, means and STDs need to be reported in the main text. The journal guidelines state "Report exact p-values wherever possible alongside the summary statistics and 95% confidence intervals. These should be reported for all key questions and not only when the p-value is less than 0.05."Have the authors corrected for multiple comparisons wherever required?

We have now reported exact p values throughout the paper. We used appropriate statistical analysis for multiple comparisons. All the comparisons were made between two samples, so they did not require multi-comparison analyses.

4) The authors report recording up to 301 cells from a single tetrode (Figure 1—figure supplement 1; including 88 cue cells and 93 grid cells; "Recordings were performed on four animals over two months."). Were repeat recordings from the same neurons on consecutive/multiple days identified and eliminated? How? If they were not eliminated, all the reported statistics suffer from inflation of degrees of freedom. Can the authors comment on this?

In Author response image 3, we have compared the original dataset to two others: 1.) a dataset in which duplicate recordings were removed using cross correlations of the real arena spatial firing rates (cell with higher maximum firing rate was kept if the correlation of the spatial firing rate in the real arena was >= 0.95) and 2.) a dataset only including cells recorded on the day when tetrodes were lowered. We found that both datasets showed consistent results with those used in the original paper. We have updated all figures and text in the paper using the dataset with duplicates removed across days (middle panel in Author response image 3).

**Author response image 3. respfig3:** Trimming the tetrode database. Cue cell sequences and scores for three different subsets of the tetrode database. Left: the original database from the original manuscript is shown. Middle: the new trimmed database with duplicates across days removed. Right: database in which cells recorded on the days in which a given tetrode was lowered are kept. In each of the three panels, the sequences of cue cells found in the database for environments 1 and 7 are shown on the left. To the right, the cue scores versus border, grid, and head direction score are shown, as well as the overall distribution of cell types.

5) What are the running speed profiles of the mice? Did they tend to slow down near the visual cues?

Animals tended to slow down only in the region of the last cue, which was associated with a water reward. Examples of the speed animals ran along the track on the last day for three tracks is shown in Figure 1—figure supplement 2.

6) "In layers 2 and 3, we consistently observed that anatomically adjacent cue cells (physical distances around 30 μm) showed more similar spatial shifts, whereas the relationship was more varied if cue cells were further apart (Figure 7G-N). The similar cue responses of adjacent cue cells suggest that they may share similar inputs or be connected."Do the cross correlations of neighbouring cells (on the same tetrode) maintain the peak at 0ms in B if they had peak at 0ms in A? If not, the two observations with tetrodes and imaging would contradict one another. In general, the claims made in G-N are rather weak. Authors should consider excluding them.The 'micro-organisation' relating physical separation to spatial shifts in responses relative to cue location is seen restricted to anatomically adjacent cue cells: is there any danger that this reflects contamination/poor localisation/diffusion of light from neighboring sources?

1) To clarify the first point about 0 millisecond correlations of tetrode data: First, we previously used a larger dataset that included some duplicate clusters from the same day and that required the removal of data at 0 msec. We have corrected for this in the paper and there are no longer any cells with a peak in the cross correlation at 0 msec. For this reason, we do not think there is a contradiction between tetrode and imaging data.

2) To clarify the second point about contamination in calcium imaging data: we did not observe the contamination of calcium responses of neighboring cells. In Author response image 4, we showed that while the anatomically adjacent cells had similar spatial shifts (same cells previously shown in Figure 7G-N), the shapes of their calcium transients were quite different, even for the transients almost occurring at the same time. These examples indicate that the similar spatial shifts of adjacent cue cells cannot be explained by contamination of their calcium signals. Upon the suggestion of the reviewers, Figure 7G-N has been removed. In addition, we also removed the original Figure 7A-F (anatomical clustering of cue cells in layers 2 and 3), because we do not consider the conclusion presented in the figure essential for the current manuscript.

**Author response image 4. respfig4:** Calcium transients of anatomically adjacent cells with similar spatial shifts. A and B are figure panels in the previous Figure 7G and H. C) Calcium responses of cell 3 (black) and cell 6 (red). Top: mean ΔF/F. Second to bottom: three examples of calcium traces (ΔF/F) showing that the shapes of calcium transients (large peaks) and the patterns of baseline activity of the two cells are different. D) Similar to C but for cell 5 and cell 2.

7) One reason for the more specific apparent correlate of firing in the VR track versus the open field might be that the viewing angle is important and this is not systematically sampled in the open field. Do the mice run in both directions on the VR – if so, do the cue cells fire at a similar cue-angle? Does this relate to the observation that place cell firing becomes more directionally modulated in VR than in real open fields (presumably because of the greater influence of vision in visually-generated VR; Acharya, Aghajan et al., 2016; Chen, King et al., 2018). In the comparison to known cell types, might these cue cells be related to landmark-vector cells (Deshmukh and Knierim, 2013) or object-vector cells (Hoydal et al., 2019), or egocentric responses recently reported in lEC (Wang et al., 2018)? Do the cue cell responses depend on the wider context – need the mouse be running (vs. passive viewing) to see firing? To what extent do cue cells fire similarly across VR environments or do they 'remap'? (this is not entirely clear from Figure 1).

We do not have data from mice running in different directions or during passive viewing but that they need to be addressed in future studies.

To address whether cue cells maintain their spatial cell type identity or remap across environments, we measured calcium responses of neurons in layer 2 of the MEC during the navigation of two different virtual tracks. We found that many cue cells showed cue-correlated responses on both tracks (Figure 6A). In general, the percentages of cue cells and non-cue cells that remained as the same cell types in two different tracks were significantly higher than chance (Figure 6B). Finally, cue cells on two tracks also showed highly correlated spatial shifts relative to cue templates (Figures 6A and C). These observations strongly suggest that the cue cell population represents cues in multiple environments in a consistent manner.

We have included a discussion of the papers that were published at the time of submission within the main paper. In this section we discuss in greater detail the papers mentioned for this reviewer comment as well as one additional paper. We have compared our results to the papers mentioned. We can include this additional discussion in the main text if necessary.

In comparison to “Egocentric coding of external items in the lateral entorhinal cortex”, (Wang et al., 2018): This paper shows a division in object representation for LEC and MEC cells. The central result of this paper is the egocentric representation of objects for cells in LEC. In this paper, the examples of “spatial non-grid cells” showed head direction preference. Some of the cells with boundary bearing sensitivity appeared to be border cells. We think cells from either of these populations could align with our cue cells, which also have spatially selective firing and head direction preference in an open arena. While it is unclear how the population of cue cells will respond to multiple cues in a complex environment, a recent study (Hoydal et al., 2019, discussed below) suggested that cue cells (object vector cells in the paper) may exhibit spatial fields around individual cues. We think the allocentric representation of items found in this paper is consistent with the known cell types within MEC. The object vector cells in the MEC also showed an allocentric vectorial representation of objects. We have not specifically investigated the allocentric or egocentric representation of our cue cells, which can be done in the future by having animals travel through the same track in different directions.

In comparison to “Influence of Local Objects on Hippocampal Representations: Landmark Vectors and Memory”, (Deshmukh and Knierim, 2013): This paper describes cells within CA1 in the hippocampus that encode spatial relationships to objects. It is possible that the landmark vector cells in this paper would have a similar response to cues in our virtual environment. However, this paper only shows responses of these cells in environments with multiple objects distributed throughout the environment. It is unclear what activity pattern these cells will exhibit in an object free, bounded arena, where we recorded our cue cells. This paper also conjectures that landmark vector cells are similar to boundary vector cells. We have shown that our cue cells do not have an obvious direct relationship to boundary vector cells. Therefore, without knowing how landmark vectors cells in CA1 respond in an object- free arena, it is unclear how related these two cell types are.

In comparison to “Entorhinal Neurons Exhibit Cue Locking in Rodent VR” (Casali et al., 2019): This paper describes cue locking of activity of cells during navigation of virtual environments. The authors note that if cues are regularly spaced then these cue locking cells will have regularly spaced firing fields that could resemble grid cell activity. They showed that these cells are not grid cells by performing experiments in real arenas. We think these cells are the same as the ones we describe in our paper.

In comparison to the unpublished paper on object-vector cells (Hoydal et al., 2019): We believe these cells belong to a similar cell population as our cue cells do since this paper and our current manuscript have made comparable findings on this cue-related cell type:

1) Similar percentages of cue/object-vector cells, with the percentage of cue cells exceeding the percentage of grid cells

2) These cells are not predominately conjunctive with other spatial cell types

3) These cells maintain cue-related firing across environments (this result has been added as Figure 6 in the current manuscript based on reviewer suggestions).

This paper did have some complementary findings/differences that our paper does not address:

1) They found that some of these cells have multiple firing fields at the location of a single object. It is possible that their objects are larger or the navigation in two dimensions changes the spatial firing field shape/number.

2) They found that these cells maintain an allocentric representation of objects, not an egocentric representation as has been observed in LEC (Deshmukh and Knierim, 2013).

In comparison to this paper, our paper provided additional information about these cue/object vector cells. We addressed the precise spike timing between simultaneously recorded cells. We also specifically studied cue cells in layers 2 and 3 of the MEC and discovered the side-preference of these cells. The fact that these cells mostly responded to single-side cues and the right cues were predominately represented in the left MEC, strongly supported a visual input-based mechanism in driving the cue cell response. For this reason, both papers provide essential information about this new cell type.

[Editors' note: further revisions were suggested prior to acceptance, as described below.]

Essential revisions:Duplicates removed database in Author response image 3, which is used throughout the paper uses spatial firing rate correlations > 0.95 as a threshold for discarding cells as being duplicates. This threshold is unreasonably high, as even stable place cells recorded in consecutive sessions in the hippocampus often have substantially lower correlation coefficients, especially in mice (e.g. Kentros et al., 1998, Figure 3). This means that the duplicates removed database is likely to still have an unreasonably high number of duplicates.The authors should show the tetrode lowering database figure shown in Author response image 3 at least as supplementary data. They must also include the tetrode lowering database stats and aggregate figures for other analyses using tetrode data, including responses to environmental perturbations, sequences, pairwise correlations etc. to convince the reader that the significance of the patterns reported is not grossly overestimated by inflation of degrees of freedom caused by the inclusion of duplicates in their dataset.

We have added figure supplements showing results for Figures 1-4 with the tetrode lowering database. We also modified the database to better eliminate duplicates by removing duplicate cells in which the firing rates in either the real or virtual environments were correlated (using the Pearson correlation, the threshold for the real arena firing rates is 0.8 and for virtual firing rates is 0.75). If either of the thresholds is exceeded then the cell with the lower maximum firing rate in the real arena is removed from the database. This reduced the database from 2107 clusters to 789 clusters; this database is now used for Figures 1-4. All results from the original paper for Figures 1-4 remained statistically significant. One figure panel only added in the previous version of the paper (first resubmission), Figure 4C, was not statistically significant for the reduced number of cells in the tetrode lowering database and has been removed.

In the revised manuscript, it is no longer clear how many animals the data were collected from, and how many neurons of different types were contributed by each animal. The animal and tetrode – wise breakdown of neurons in the tables included in the previous version are essential. Tables showing number of units of different kinds recorded from each tetrode in each animal have been eliminated from Figure 1—figure supplement 1. They should be put back, with numbers for both duplicates removed database as well as for tetrode lowering database.

These changes have been made to Figure 1—figure supplement 1.

Related to this, it is not clear how many animals contributed to the new data shown in the new Figure 7. Hence, it is impossible to figure out if the reported results are reproducible across animals. Please mention number of animals included for each analysis/figure in the Results.

The information about the number of animals for each result has been included for all figures.

"In Region A, there was a spread in the temporal shifts for pairs of cue cells and these temporal shifts were correlated for the two tracks (Figure 5C left, Pearson correlation = 0.52, p=9×10-5). However, the temporal shifts in Region B of the two tracks were less correlated: while a similar spread of temporal shifts was observed when cue cells were recorded on the with cues track (plotted along the x-axis of the bottom right panel in Figure 5C), most cue cell pairs did not have a correlated phase in the relative spike timing when cues were missing (plotted along the y-axis of the bottom right panel in Figure 5C right, “correlation not significant”). The fact that the spike timing relationship between cue cell pairs is maintained only when cues are present suggests that these cells are driven to be active in a sequential manner by locomotion past cues."Correlation coefficient and p value for region B needs to be included, as requested in the previous review – stating "correlation not significant" is not sufficient. Furthermore, to make the claim that their data suggests that "cells are driven to be active in a sequential manner by locomotion past cues", the authors should demonstrate that the slopes in region A and B shown in Figure 5C are significantly different from one another.

We decided to remove this figure since it does not add to the main finding of this paper that cue firing fields are no longer present when cues are removed. The sequential nature of the averaged activity of the population of cells is evident in Figures 4 and 5 by direct inspection.